# Masks Can Be Distracting:
# On Context Comprehension in Diffusion Language Models

**Julianna Piskorz** [1] **Cristina Pinneri** [2] **Alvaro H. C. Correia** [2] **Motasem Alfarra** [2] **Risheek Garrepalli** [2]
**Christos Louizos** [2]

## Abstract

Masked Diffusion Language Models (MDLMs) have recently emerged as a promising alternative to Autoregressive Language Models (ARLMs), leveraging a denoising objective that, in principle, should enable more uniform context utilisation. In this work, we examine the context comprehension abilities of MDLMs and uncover two key limitations. First, despite their more global training objective and bidirectional attention mechanism, similarly to ARLMS, **MDLMs exhibit a strong locality bias**: performance is highly sensitive to the position of relevant information within the input, favouring local over distant context. Second, appending a large number of **mask tokens–required for generation–can significantly degrade context comprehension** in models trained from scratch. Through systematic ablations, we find that these masks **act as distractors**, reducing the model's ability to process relevant information. To address and further study this undesirable behaviour, we introduce the mask-agnostic loss function that encourages predictions to remain invariant to the number of appended masks. Fine-tuning with this objective substantially mitigates the distracting effect of masks, improving robustness of MDLMs. Overall, our findings reveal critical limitations of the current MDLM training paradigm, with implications for training, evaluation and deployment.

## 1. Introduction

Diffusion Language Models (DLMs) have recently emerged as a promising alternative to autoregressive models (ARLMs), offering parallel generation and bidirectional context modelling through iterative denoising (Austin et al., 2021; Lou et al., 2024). Within this family, masked DLMs (MDLMs) (Sahoo et al., 2024; Shi et al., 2024) have seen rapid progress, scaling competitively and showing strong perplexity and speed on standard benchmarks (Nie et al., 2025; Song et al., 2025). Yet, despite these advances, it remains unclear how MDLMs use context during inference, and to what extent their distinct training objective alters well-known inductive biases of ARLMs (Liu et al., 2024; Barbero et al., 2024). In this work, we present a systematic study of context comprehension in MDLMs and uncover limitations that have immediate implications for how these models are trained, evaluated and deployed.

Our analysis reveals that although their masked diffusion objective permits more global context processing in principle, context use is far from uniform. MDLMs exhibit strong locality effects, relying heavily on nearby context rather than uniformly integrating information across the input sequence. Moreover, generation-time design choices such as the number and placement of mask tokens can significantly alter performance, raising fundamental questions about the robustness of MDLMs and the validity of current evaluation practices.

This sensitivity traces back to a core design element of MDLMs: the use of mask tokens both during training (via randomised masking in the denoising objective) and at inference (to delimit the prediction span). While intended as neutral scaffolding, we discover that these tokens can disrupt context processing, acting as distractors. Our analysis uncovers a striking inverse scaling law in MDLMs trained from scratch using the masked diffusion objective: as more masks are appended, performance drops significantly. We conduct careful analysis of this degradation, testing alternative hypotheses for its origins. We show that the degradation in behaviour is not just quantitative: it is also reflected in context processing patterns, levelling performance across different locations of input, but at a consistently lower base-

---

[1]University of Cambridge, United Kingdom. Work done during an internship at Qualcomm. [2]Qualcomm AI Research. Qualcomm AI Research is an initiative of Qualcomm Technologies, Inc.. Correspondence to: Julianna Piskorz <jp2048@cam.ac.uk>.

line. Overall, we discover that the very tokens meant to guide generation end up clouding it. Far from passive place-holders, *masks can be distracting*, and their influence should be treated as a crucial factor in the *design, evaluation, and deployment* of MDLMs.

In this work, using systematic empirical analysis, we make the following contributions towards identifying and explaining the fundamental limitations of MDLMs:

- **Locality bias (Section 4)**: We provide the first systematic evidence that MDLMs exhibit a strong *locality bias*, prioritising information near the masked token despite their global denoising objective.
- **Inverse scaling with masks (Section 5)**: We uncover an *inverse scaling law with extra masks*: in MDLMs trained from scratch using the masked diffusion objective, additional mask tokens can significantly degrade performance, especially in long-context settings.
- **Mask-agnostic fine-tuning (Section 6)**: We propose a *mask-agnostic objective* that enforces prediction invariance to mask count, improving robustness of MDLMs.
- **Evaluation guidelines (Section 7)**: We establish mask configuration as a critical factor in MDLM evaluations and recommend standardised practices for fair benchmarking.

## 2. Related Works

**Diffusion Language Models.** Diffusion Language Models (DLMs) have recently emerged as a promising alternative to the dominant autoregressive paradigm for text generation (Nie et al., 2025; HKU NLP Group; Song et al., 2025; Khanna et al., 2025). Unlike GPT-style models that generate text sequentially, DLMs employ an iterative denoising process that starts from a noisy representation and progressively reconstructs coherent text, enabling parallel token generation and bidirectional context modelling. While early research explored both continuous and discrete diffusion formulations for text, the discrete masked diffusion objectives (Sahoo et al., 2024; Lou et al., 2024; Austin et al., 2021; Shi et al., 2024; Ou et al., 2025) have recently dominated the landscape, allowing DLMs to effectively scale to larger model sizes and achieve competitive perplexity on standard benchmarks (HKU NLP Group; Nie et al., 2025). MDLMs have attracted a lot of attention for their potential to speed up inference (Kim et al., 2025; Frans et al., 2025; Song et al., 2025; Israel et al., 2025; Wu et al., 2025; Park et al., 2025; Agrawal et al., 2025) and improve controllability (Rector-Brooks et al., 2025; Gaintseva et al., 2025; Pani et al., 2025). However, to the best of our knowledge, a comprehensive evaluation of the influence of the masked diffusion training objective on the models' context comprehension abilities is still missing.

**Context Comprehension in Language Models.** Language models do not process information provided in the input uniformly (Sun et al., 2021; Qin et al., 2023; Barbero et al., 2024). Two well-documented position biases are primacy bias—a tendency to favour information appearing early in the input—and recency bias, where information near the end is weighted more heavily. These effects combine to produce the characteristic U-shaped accuracy curve in autoregressive models, often referred to as the *lost-in-the-middle* phenomenon (Liu et al., 2024). Empirical evidence for these biases comes from variations of the needle-in-the-haystack experiments (Kamradt) and related benchmarks across diverse tasks, including information retrieval (An et al., 2024), multi-document question answering (Liu et al., 2024), graph reasoning (Firooz et al., 2024), and in-context learning (Kossen et al., 2024). Primacy bias has been often attributed to the causal attention mask (Barbero et al., 2024), while recency bias has been linked to the training data distributions and the next-token prediction objective (Sharan et al., 2018; Barbero et al., 2024; An et al., 2024).

Whether MDLMs–trained on similar text corpora but with a fundamentally different objective–exhibit comparable position biases remains an open question. Prior work has explored related but distinct aspects of MDLMs: Liu et al. (2026) evaluated LLaDA on needle-in-the-haystack tasks to study *generalization to unseen context lengths*, but their setup was too simple to reveal recency effects within the models' context lengths. Similarly, Shansan et al. (2025) examined the "AR-ness" of MDLMs, defined as a *preference for left-to-right decoding*. In contrast, our work investigates whether MDLMs display AR-like tendencies in *processing* the context, rather than in their decoding strategy.

## 3. Experimental Setup

We focus our analysis on open-source MDLMs, to have full control over generation settings: we compare the performance of **LLaDA-8B** with Llama3-8B (AI@Meta, 2024) (an ARLM with a similar architecture) and **Dream-7B** against Qwen2.5-7B (Yang et al., 2024; Team, 2024) (an ARLM model used to initialise Dream-7B). To improve the generalisability of our findings, we provide additional results on **LLaDA-MoE** (Zhu et al., 2025) in Appendix A.1. We note that while LLaDA was *trained from scratch* using the masked diffusion loss (Sahoo et al., 2024), Dream was *initialised from ARLM weights*, thus constituting an interesting interpolation point between ARLMs and and LLaDA. Since we prioritise accuracy over generation diversity, we use greedy decoding for all models we consider.

The positional bias in LMs has typically been studied using information retrieval tasks, such as multi-document question answering tasks (Liu et al., 2024) or needle-in-a-haystack tasks (Kamradt; An et al., 2024). However, these tasks are relatively easy (An et al., 2024), and in order to showcase

the positional bias of the models, they require significantly larger context lengths than those of the MDLMs we consider in this work (LLaDA and Dream were trained using context lenghts of 4096 and 2048 tokens respectively). For instance, Liu et al. (2026) have demonstrated that LLaDA-8B-Base achieves 100% retrieval accuracy on the Needle-In-A-Haystack tests (Kamradt) within the context length that it has been trained on. Thus, as in this work we are *not* interested in studying generalisation to larger context lengths, we depart from traditional information retrieval setups.

Instead, we propose a suite of few-shot learning tasks allowing to evaluate the sensitivity of LMs to information placement, while remaining within the context limits of all the studied models. The few-shot learning tasks we consider are inspired by Todd et al. (2024) and take the form of multiple-choice questions, requiring the model to infer an abstract rule from examples. For example, given three words, the model needs to choose the one which is an adjective, which would be formatted as:

```
Options: (A) knit, (B) quirky, (C) persuade
Answer:[B].
```

We purposefully design the tasks so that *the correct answer spans just a single token*, to enable robust evaluation of context processing abilities through the accuracy metric (avoiding relying on more ambiguous metrics such as generative perplexity (Zheng et al.)), and allowing for a more fine-grained analysis of the model's behaviour (gradient analysis, entropy of predictions) without having to control for the effects of token decoding order. The resulting tasks span only a small number of tokens, thus allowing us to remain within the context length of all the studied models.

To be able to manipulate the placement of relevant information within the context, alongside our selection of 8 relevant *word* tasks (e.g. choose adjective, verb, fruit), we also construct 2 distractor tasks based on *numbers* instead (e.g., choose the largest or smallest number). Considering all combinations of relevant and distractor tasks leads to a suite of 16 tasks which we use for evaluation, each containing 1000 test points. Crucially, the tasks are constructed so that the governing rule can only be learned through exposure to *multiple* examples. Thus, by randomly mixing the relevant and irrelevant examples, we can test the model's ability to comprehend long contexts. *Unless otherwise stated, in our figures the shaded regions mark 95% confidence intervals, computed over the 16 datasets contained in our suite of few-shot learning tasks.*

We also present results on other datasets in Appendix A: **HotPotQA** (a multi-hop reasoning dataset), **GSM8k** (a dataset of grade-school maths word problems) and **a multi-dimensional classification dataset**. Additional experimental details are provided in Appendix C.

## 4. Are MDLMs Location-Sensitive?

**Motivation.** Autoregressive language models (ARLMs) dominate the current text generation paradigm, but their next-token prediction objective enforces a strict left-to-right generation order, introducing locality biases–such as recency bias–that hinder effective use of long-range context (Sun et al., 2021; Liu et al., 2024; Kossen et al., 2024; Qin et al., 2023). These biases are widely attributed to the autoregressive loss, which inherently prioritises recent tokens due to its sequential structure (An et al., 2024; Barbero et al., 2024; Bachmann & Nagarajan, 2024; Sharan et al., 2018). This raises a key question: if we move away from the AR objective, can we reduce these biases? Masked Diffusion Language Models (MDLMs) offer a compelling opportunity to explore this hypothesis. Unlike ARLMs, MDLMs optimise a masked diffusion objective that denoises tokens *across the whole sequence* in parallel, at different noise levels, potentially enabling more global context integration. Moreover, the masked diffusion objective has been shown to be equivalent to any-order autoregressive modelling (Shuchen et al., 2025), suggesting that MDLMs might overcome positional constraints inherent to ARLMs. We investigate whether the diffusion objective truly mitigates locality biases and improves long-context comprehension or if such biases persist despite the training paradigm shift.

### 4.1. Is the Performance of MDLMs Sensitive to the Location of Relevant Information?

**Setup.** To assess whether model performance depends on the position of relevant information, we systematically vary the location of the *relevant* in-context learning examples within the prompt and measure the resulting accuracy on test questions. Specifically, we use 10 relevant examples (grouped together into one block), and 40 distractor examples. We keep the order of examples within the relevant and distractor groups fixed across all conditions, varying only the position of the relevant block within the overall sequence. We put the test example at the right end of the provided context, in an auto-regressive fashion.

**Results.** Figure 1 summarises the effect of information placement on model accuracy. Despite being trained with a masked diffusion objective–which does not enforce a strictly sequential prediction order–both MDLMs exhibit strong sensitivity to the position of relevant examples. Performance is highest when relevant information appears immediately before the test question, indicating a significant **recency bias**. Unlike ARLMs, which often display a U-shaped pattern (high accuracy when relevant examples are at the beginning *or* end of the prompt) (Liu et al., 2024; Barbero et al., 2024), MDLMs show a monotonic decline in accuracy as relevant information moves farther away. We do not observe a strong primacy effect in MDLMs, which aligns with expectations,

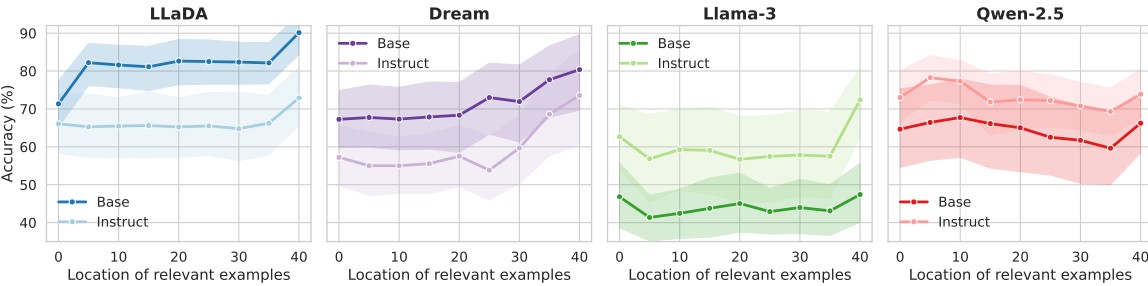

*Figure 1.* **MDLMs display a recency bias.** The performance of both MDLMs (LLaDA and Dream), as well as the studied ARLMs, is sensitive to the placement of relevant information within the context. For MDLMs, the performance degrades significantly when the relevant information is placed far away from the test question, suggesting a recency bias.

as the primacy effect has been attributed primarily to the causal attention mechanism (Barbero et al., 2024).

### 4.2. Does 'Locality' Depend on Mask Placement?

**Setup.** The previous experiment revealed a strong recency bias in MDLMs, but it did not clarify its origin: does the bias arise because models generally prioritise information near the right edge of the context, or because they attend most strongly to the region around the mask token? To disentangle these factors, we repeat the previous experiment while varying the position of the test question (with its answer masked) within the prompt.

**Results.** Figure 2 shows that across all settings, model performance is highest when relevant information is placed *near* the masked question. This indicates that the previously observed recency bias is, in fact, a broader **locality bias**: MDLMs prioritise information close to the prediction target, regardless of its absolute position in the prompt. Interestingly, performance is consistently lowest when the masked question appears at the beginning of the input. We also note that for Dream, the performance is generally better when the relevant information is located *to the left* of the mask–suggesting a left-directed bias that resembles the behaviour of ARLMs that Dream was initialised from.

What is the source of the locality bias? Although MDLMs are trained on a more delocalised decoding objective than ARLMs, the masked diffusion loss is scaled by $1/p$, where $p$ is the probability of masking a token (Nie et al., 2025; Sahoo et al., 2024). Consequently, training places greater weight on cases where only few tokens are masked–scenarios where nearby context is usually sufficient for prediction, as in next-token prediction setting (Sharan et al., 2018). We hypothesise that this encourages MDLMs to rely on nearby context when processing the inputs.

### 4.3. Quantifying the Bias with Gradient Analysis

**Setup.** To deepen our understanding of the locality bias in MDLMs and ARLMs, we perform gradient attribution

analysis (Lopardo et al., 2024), which quantifies how sensitive the model's prediction is to changes in each input token. Specifically, we compute the L2 norm of the gradients of the logit corresponding to the predicted answer token with respect to the input token embeddings. This provides a mechanistic measure of each token's influence on the output, which has been shown to carry more signal for explaining the model behaviour than naive attention analysis (Lopardo et al., 2024).

We use a dataset containing 10 relevant examples and 40 distractors, randomly mixed together to ensure the model must process the entire context to arrive at the answer. While the examples remain fixed across runs, their relative ordering is randomised across 30 seeds. If the models were location-invariant, gradient magnitudes would be roughly uniform across the positions. As the in-context examples do not change for different test questions, for computational efficiency we evaluate the gradients for a sample of 20 test questions for each task only.

**Results.** Figures 17, 18, and 20 show *normalised* gradient scores across different in-context examples and mask placements. Both ARLMs and MDLMs exhibit non-uniform attribution patterns, with the local context affecting the output most strongly. In contrast to Figure 1, where MDLM performance degraded monotonically, the gradient attribution plots in Figure 17 exhibit the characteristic U-shape associated with both recency and primacy effects across all models. However, MDLMs–particularly the base models–display more evenly distributed gradient values than ARLMs, suggesting greater potential for global context utilisation.

> **Takeaways.** Although MDLMs are trained with a masked diffusion objective that denoises tokens across the entire sequence in parallel, they still exhibit a strong **locality bias**: performance depends heavily on the proximity of relevant information to the masked question, with local information processed more effectively.

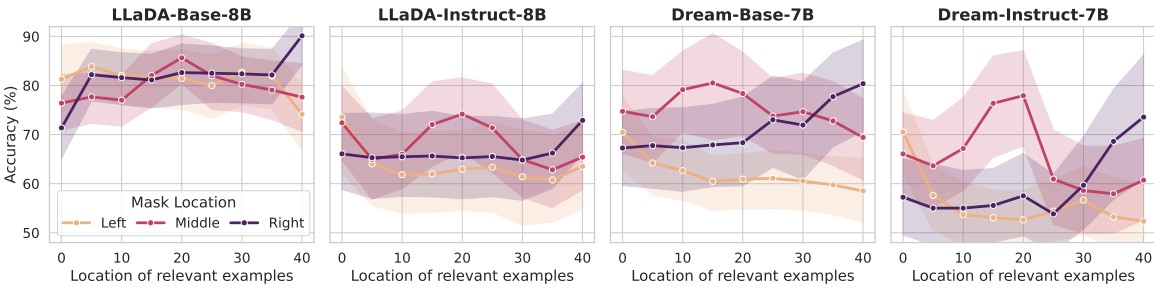

*Figure 2.* **MDLMs prioritise information placed closest to the mask.** All studied MDLMs perform best when the relevant information is placed near the masked token, regardless of the position of the test question.

## 5. The Distracting Effect of Extra Masks

**Motivation.** The previous section established that MDLMs do not process context uniformly; instead, they prioritise information closest to the mask. However, our analysis so far has been restricted to single-token answers, for which we allocated *a single mask token* during decoding. To gain a more comprehensive understanding of MDLMs' context comprehension, we now investigate how their performance changes when additional masks are introduced. As mask tokens are a unique feature of MDLMs, *their impact on context processing has not been systematically studied before*. Our hypothesis is that increasing the number of masks may reduce the model's local focus, encouraging it to integrate information more globally across the context.

### 5.1. Do Extra Masks Affect MDLMs' Performance?

**Setup.** To measure how additional mask tokens affect MDLMs' context comprehension, we append varying numbers of mask tokens to the input prompt. We use 10 relevant and 40 distractor examples, mixing these two groups randomly together, to force the model to process the entire input context. In our format, the first mask token always corresponds to the answer for the test question. We decode the entire sequence in a single step but evaluate only the prediction for this first mask, ignoring all others. This setup isolates the effect of extra masks on the model's ability to correctly predict the target answer token, without introducing confounding factors from multi-step decoding (we study the effect of decoding the extra masks sequentially in Section 5.5).

**Results.** Contrary to our initial hypothesis–that additional masks might improve global reasoning–we observe a consistent performance degradation as the number of masks increases (Figure 3). This trend holds for both LLaDA-Base and LLaDA-Instruct (as well as for LLaDA-MoE models, as can be seen in Figure 12). This is a surprising result: while extra masks could be expected to increase prediction entropy, inducing high uncertainty in generations, it is worrisome that for LLaDA models they lead to a consis-

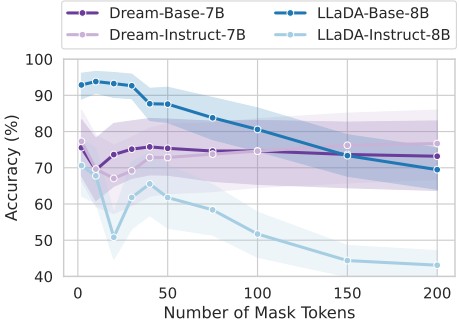

*Figure 3.* **Performance of LLaDA decreases significantly with added mask tokens,** while Dream is more robust.

tent, monotonic degradation in accuracy even under greedy decoding—implying that with extra masks the mode of the model's token distribution is actively shifting to incorrect answers. For the base model, one plausible explanation is a distribution shift: during training, random noising rarely produces long unmasked prefixes followed by large contiguous mask spans. However, this scenario is not unusual for the instruct model, which exhibits a similar decline. Similar detrimental effect of too large numbers of masks was also seen in concurrent works (Li et al., 2025), although to a lesser extent.

Dream models appear more robust to large numbers of masks, but still exhibit a noticeable drop (6 and 8 percentage points for the base and instruct model, respectively) when approximately 20 masks are added, indicating that they are not fully invariant to the extra masks. We hypothesise that this difference may stem from Dream models being initialised from the weights of the autoregressive Qwen-2.5, making mask tokens *less integral* to their architecture and training dynamics. In the following sections, we study this phenomenon of performance degradation due to extra masks in more detail, analysing several possible hypotheses which could explain this behaviour.

### 5.2. Do MDLMs Get Equally Distracted on All Tasks?

**Setup.** We begin our analysis by investigating whether performance drop caused by extra masks is linked to impaired context comprehension. To that end, we examine

| Model Name | Masks | Non-Masks (last 50) | Non-Masks |
|---|---|---|---|
| Dream-Base | 0.282 ± 0.040 | 0.012 ± 0.007 | 0.005 ± 0.003 |
| Dream-Instruct | 0.144 ± 0.031 | 0.030 ± 0.005 | 0.018 ± 0.002 |
| LLaDA-Base | 0.234 ± 0.021 | 0.005 ± 0.002 | 0.005 ± 0.002 |
| LLaDA-Instruct | 0.220 ± 0.031 | 0.057 ± 0.014 | 0.017 ± 0.003 |

*Table 1.* **MDLMs are particularly sensitive to mask tokens.** We show the average normalised gradients attributed to the mask tokens, compared to all the other tokens in the input sequence.

how the effect of extra masks changes as the context length required to solve the task increases. Specifically, we vary the number of distractor examples in the prompt while keeping the number of relevant examples fixed, mixing the two groups together randomly. If extra masks indeed disrupt context processing, *we expect their negative impact to grow as more distractors are added*. This is because the model must filter relevant from irrelevant information over a longer context, and extra masks may disrupt its attention allocation.

**Results.** Figure 4 shows that for LLaDA, performance degradation due to extra masks generally increases with the number of distractors, and thus with the effective context length. This suggests that additional masks impair the model's context processing abilities.

We provide further evidence for this claim in Appendix A.3. There, we compare the degree of performance degradation caused by extra masks with the gains achieved when increasing the number of in-context examples across different tasks. We find a strong correlation: tasks that benefit most from additional context are also the most vulnerable to mask-induced degradation. This reinforces the conclusion that extra masks inhibit long-context comprehension.

### 5.3. Is the Distracting Effect Reflected in the Gradients?

**Setup.** To further mechanistically assess how strongly MDLMs prioritise the extra mask tokens over any other tokens in the input, we analyse gradient-based attributions. Using the dataset configuration from Section 4.3, we append 50 mask tokens to the input and measure the normalised gradient of the masked answer token (i.e., the first mask) with respect to all other tokens in the sequence. This quantifies the influence of the added masks on the model's prediction and the model's sensitivity to their presence relative to the surrounding context.

**Results.** Table 1 reports the average normalised gradients for three token groups: (i) the added mask tokens, (ii) the last 50 non-mask tokens closest to the target mask, and (ii) all non-mask tokens. Across all models, gradient magnitudes attributed to mask tokens are markedly higher than those attributed to either non-mask group. This pattern indicates that MDLMs are disproportionately affected by the added masks. While Dream models seem to be mechanistically strongly affected by the masks, this is not reflected in their performance: suggesting that the differences in the training

paradigm might lead to different inductive biases not fully reflected in the gradient analysis alone. We also note that the last 50 non-mask tokens (i.e. the 50 tokens located directly to the left of the mask) have higher gradient scores than non-mask tokens in general, reiterating the recency bias.

### 5.4. Is the Degradation Caused by Repeated Tokens?

**Setup.** In the previous sections, we hypothesised that extra masks degrade performance because they act as distractors, drawing attention away from relevant context. To further validate this hypothesis and rule out alternative explanations, we test whether the performance degradation observed earlier is caused by the presence of the mask tokens specifically– rather than simply by appending many identical tokens. To evaluate this, we repeat the experiment from Section 5.1 but replace the extra masks with a relatively neutral token sequence: the string " . " repeated multiple times. This ablation allows us to isolate the effect of mask tokens and verify that the observed behaviour is not merely due to an out-of-distribution repetition.

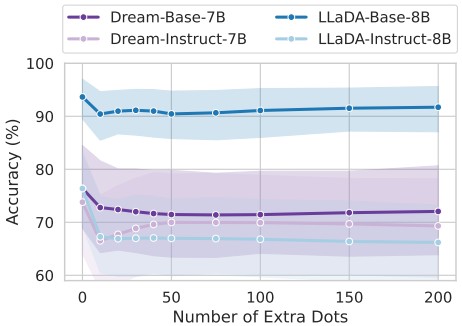

*Figure 5.* **Extra dots do not degrade performance as strongly as extra masks.**

**Results.** Figure 5 shows that appending extra dots to the input has only a minor impact on the performance of LLaDA, especially compared to the substantial degradation seen in Figure 3 (for the dots, performance decreases by up to 3 and 10 percentage points for the base and instruct models respectively, compared to 23 and 27 percentage points for the masks). This confirms that in LLaDA the performance drop is driven by the presence of *masks* specifically, rather than by the mere repetition of identical tokens. For Dream, the effect of the masks and the dots are largely similar.

### 5.5. Can the Negative Effect be Fixed by Unmasking?

**Setup.** We examine whether the degradation caused by extra masks can be alleviated at inference time via iterative unmasking, that is, progressively resolving masked positions. This procedure is consistent with the denoising paradigm which MDLMs are trained for, where generation typically proceeds by unmasking the entire sequence in multiple steps. We run 40 decoding steps and compare two

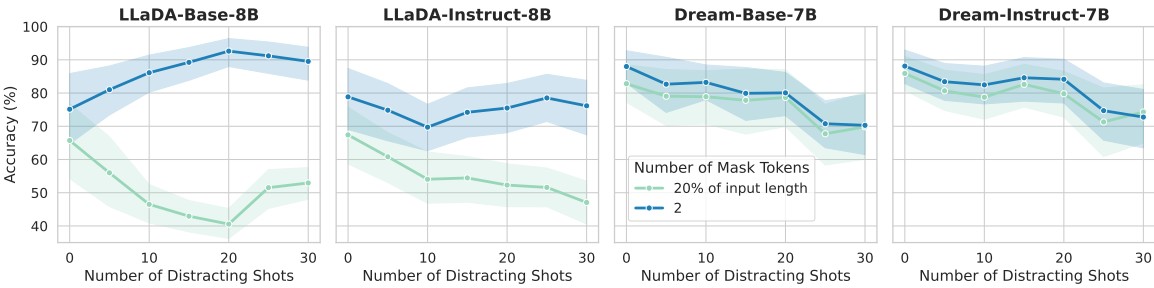

*Figure 4.* **For LLaDA, performance degradation becomes more significant as the context length increases.** We do not observe a similar effect for Dream, which is robust to the effect of extra masks.

selection strategies for unmasking: choosing which tokens to unmask at random or according to the highest confidence.

**Results.** Figure 6 shows that unmasking (with 40 steps) markedly improves accuracy, recovering the performance lost due to the extra masks. This is especially true for the high-confidence unmasking strategy. This corroborates our findings in Section 5.4: extra masks act as strong distractors and removing them, even with imperfect generations, restores focus on relevant context. While effective, this approach adds latency as it requires multiple decoding passes, which might not be desirable for specific hardware-constrained applications.

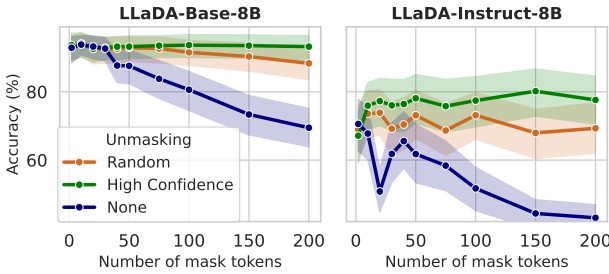

*Figure 6.* **Unmasking recovers accuracy lost to mask-induced distraction**. Unmasking strategies improve performance compared to no unmasking (None), with High Confidence consistently out-performing Random, especially as the number of masks increases.

### 5.6. Do Extra Masks Affect the Locality of the MDLMs?

**Setup.** Finally, we revisit the question that motivated us to explore the effect of masks: can additional masks alter the locality bias observed in MDLMs (Section 4)? To test this, we repeat the experiment from Figure 1 but append varying numbers of extra masks to the input. This allow us to assess how extra masks influence the model's ability to use information at different positions within the prompt.

**Results.** The results shown in Figure 7 validate our original hypothesis: as we add more masks, the performance indeed becomes more uniform across positions. However, this uniformity mainly reflects consistently poor results.

> **Takeaways.** The context comprehension abilities of MDLMs can be severely impaired by the presence of **extra mask tokens** in the input. This degradation worsens as context length increases, suggesting that masks act as strong **distractors** which can interfere with long-context processing. Controlled experiments confirm that this effect is specific to mask tokens (rather than a byproduct of repeated tokens only), and can be largely rectified by unmasking. We further find that the extra masks diminish the locality bias, reducing the location-dependent variation in performance.

## 6. Reducing the Distracting Effect Through Mask-Agnostic SFT

**Motivation.** In many practical settings, asking the user to specify the expected token length of a valid answer a priori may be challenging. Therefore, we see robustness to variations in the number of mask tokens as a desirable property for MLDMs. Existing models can recover such robustness at additional inference cost (see Figure 6), but we seek alternative approaches that remain effective even with few decoding steps. Such methods would benefit low-latency applications and could improve inference-time decoding strategies such as adaptive parallel decoding (Israel et al., 2025), aiming to generate multiple tokens in parallel.

To this end, we propose a supervised fine-tuning scheme that promotes prediction invariance with respect to the number of masks appended at the end of the context, encouraging the model to focus on the core task rather than auxiliary mask structure. Crucially, our fine-tuning scheme also serves as an additional, mechanistic tool for validating that the observed performance degradation is the result of extra masks having a *distracting effect*—we show that teaching the model to disregard the extra masks at generation can recover performance.

### 6.1. Formulation of the Mask-Agnostic Loss Function

To encourage invariance to the number of extra masks, we propose a **mask-agnostic (MA) loss**. Consider

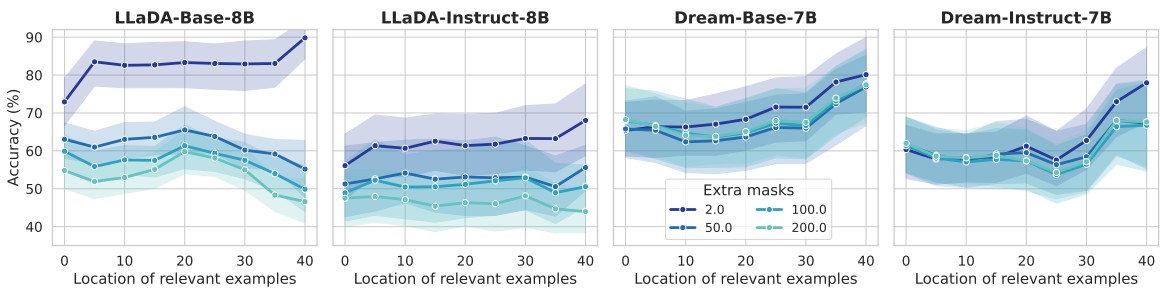

*Figure 7.* **Extra masks diminish the locality bias.** We measure performance sensitivity to the location of relevant information as extra masks are added. With more masks, accuracy becomes less location-dependent, mainly because it declines across all positions.

prompt–answer pairs, where $\boldsymbol{q} = (q^1, \ldots, q^{n_q})$ is the tokenised prompt and $\boldsymbol{a} = (a^1, \ldots, a^{n_a})$ is the tokenised answer. We construct a noised version of the answer, $\tilde{\boldsymbol{a}} = (\boldsymbol{1} - \boldsymbol{u}) \circ \boldsymbol{a} + \boldsymbol{u} \circ \boldsymbol{m}$, where $\boldsymbol{1} \in \mathbb{R}^{n_a}$ is the vector of 1s, $\boldsymbol{m} \in \mathbb{R}^{n_a}$ is the vector of mask tokens, and $\boldsymbol{u} = (u^1, \ldots, u^{n_a})$ is a vector of samples from a Bernoulli distribution $(u^1, \ldots, u^{n_a} \overset{\text{iid}}{\sim} Ber(p))$ for masking probability $p$. Here, $\circ$ denotes element-wise vector multiplication.

Let $\boldsymbol{q} \oplus \tilde{\boldsymbol{a}}$ denote the concatenation of the prompt and the noised answer tokens. To compute our MA loss, we construct two alternative versions of this input, with different numbers of mask tokens appended. That is, we select $l_1, l_2 \in \mathbb{Z}$ randomly without replacement from the range $[0, N - (n_a + n_q)]$, where $N$ is some pre-defined maximum context length. We then construct two inputs: $\boldsymbol{x_1} = \boldsymbol{q} \oplus \tilde{\boldsymbol{a}} \oplus (m \otimes l_1) = (x_1^1, \ldots, x_1^{n_q+n_a+l_1})$ and $\boldsymbol{x_2} = \boldsymbol{q} \oplus \tilde{\boldsymbol{a}} \oplus (m \otimes l_2) = (x_2^1, \ldots, x_2^{n_q+n_a+l_2})$. The corresponding labels (not noised) are: $\boldsymbol{x} = \boldsymbol{q} \oplus \boldsymbol{a} = (x^1, \ldots, x^{n_q+n_a})$. Further, let $\mathcal{A}$ denote the set of indices of the elements of $\boldsymbol{x_1}$ and $\boldsymbol{x_2}$ which correspond to the answer-part of the input. With this notation in hand we can define our loss as follows:

$$\mathcal{L}_{CE} = -\frac{1}{2pn_m} \sum_{i=1,2} \sum_{j \in \mathcal{A}} \mathbb{1}\{x_i^j = m\} \log p_\theta(x^j | \boldsymbol{x_i}),$$

$$\mathcal{L}_{TV} = \frac{p}{n_m} \sum_{j \in \mathcal{A}} \mathbb{1}\{x_1^j = m\} TV\left(p_\theta(x^j | \boldsymbol{x_1}), p_\theta(x^j | \boldsymbol{x_2})\right),$$

where $p_\theta$ is the MDLM distribution and $n_m = \sum_{j \in \mathcal{A}} \mathbb{1}\{x_i^j = m\}$ is the number of masked tokens. Our final MA-loss is then constructed as $\mathcal{L}_{MA} = \alpha \mathcal{L}_{CE} + \beta \mathcal{L}_{TV}$ for scaling parameters $\alpha$ and $\beta$.

The first term (**CE loss**) is a cross-entropy loss on the generated answer, ensuring that the model's predictions match the ground-truth answers regardless of how many additional masks are appended. We scale this term by $1/p$, following the standard masked diffusion objective (Sahoo et al., 2024; Nie et al., 2025). The second term (**TV loss**) is a total variational distance that explicitly encourages the probability distributions of the answer tokens to remain consistent across different masking configurations. We scale this term by $p$ to ensure that the distributions are aligned even when

there are scarcely any unmasked tokens in the answer. As we explain in Section 4.2, in this case the generations are less constrained by the neighbouring tokens, and thus similarity under different masking conditions is crucial to ensure robustness. We further divide both terms by $n_m$ to ensure that loss is calculated on a per-token basis (to account for the possible large variations in the answer lengths, and hence in the number of masked tokens per input). We provide a pseudo-code for the loss in Appendix B.

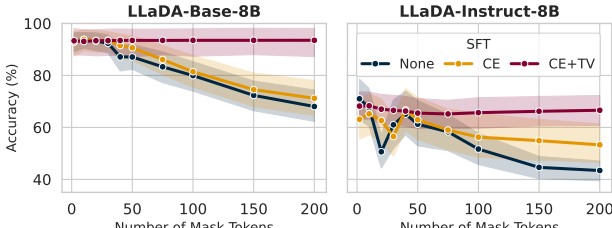

*Figure 8.* **MA loss rectifies the effect of extra masks.**

### 6.2. Experimental Validation

**Training Details** To empirically verify the effectiveness of our MA loss function, we fine-tune LLaDA models using LoRA adapters. We also train an ablated model with the CE loss only, setting $\beta = 0.0$. We train on a subset of the OpenOrca dataset (Lian et al., 2023), for $\approx 1.2$k gradient descent steps. OpenOrca is an instruction-tuning dataset, *not matching our ICL evaluation setup*. Thus, its diversity ensures that fine-tuning induces more global changes in the model, rather than overfitting to the ICL-specific task structure. Training details are in Appendix B.

**Does the MA Loss Rectify Performance Degradation Caused by Extra Masks?** Figure 8 shows that fine-tuning both LLaDA-Base and LLaDA-Instruct model with our MA loss allows to improve the performance of the models, making them more robust to variations in the number of masks appended to the input. Similar effects are visible in the LLaDA-MoE-Base (Appendix Section A.1). The CE loss on its own does not have a similar effect, emphasising the importance of regularising generation with the TV loss directly. In Figure 26 we also show the effect of MA loss on the logits of the model, showing that our SFT procedure

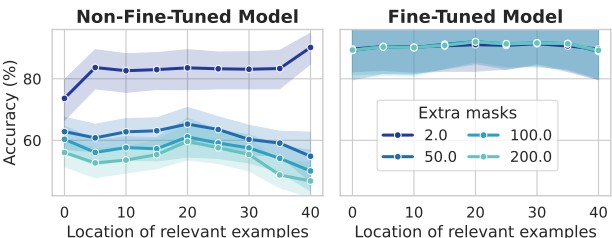

*Figure 9.* **MA loss reduces the locality bias** of LLaDA-Base.

reduces the entropy of the model and makes it significantly smoother as a function of masks, thus increasing robustness. Unlike unmasking (Section 5.5), which recovers accuracy through multiple decoding passes, our approach achieves similar robustness in only **a few decoding step** (we show performance improvements when using for 2, 4 and 6 decoding steps in Appendix A.4). This makes it attractive for low-latency applications and for distillation pipelines, where minimising generation steps is critical for efficiency and model compression. In Appendix A.5 we further verify that the MA loss *does not* degrade the language modelling performance of the fine-tuned models.

Further, the success of our mask-agnostic loss offers additional insights into the nature of the sensitivity of LLaDA models to the extra masks: it proves that this behaviour is not an insurmountable architectural flaw, but rather a training artifact, which *can be corrected* by enforcing invariance to the number of mask tokens.

**Does the MA Loss Affect the Locality of the Model?** We show in Figure 9 that the MA loss has the added benefit of reducing the locality bias of the LLaDA models. We hypothesise that by teaching the model to ignore the extra masks, we encourage it to pay more attention to the provided context, thus improving context comprehension. This further emphasises the inherent link between the extra masks and context comprehension abilities. We provide corresponding results for the LLaDA-Instruct model in Appendix A.6.

> **Takeaways.** We introduced the **mask-agnostic loss**, a fine-tuning objective that encourages invariance to the number of masks appended during generation. We show that this strategy can improve robustness with only a few decoding steps while reducing the locality bias, emphasising the inherent link between the extra masks and context comprehension abilities.

## 7. Discussion, Conclusions and Future Work

**The "Mask Tax" on Parallel Decoding.** We believe our findings uncover a critical friction point for the practical deployment of MDLMs, specifically regarding their propensity for fast (parallel) decoding. One of the primary appeals of MDLMs is accelerating inference by generating many tokens simultaneously, which requires initialising the input

with a large number of mask tokens. It is widely acknowledged that decoding in fewer steps can degrade performance because the model neglects dependencies between generated tokens. However, our work identifies a second, distinct source of degradation. We show that the mere presence of the large number of mask tokens—required for parallel generation—actively acts as a distractor, impairing context comprehension before token dependencies even come into play. Acknowledging this "mask tax" is crucial for designing more robust fast samplers.

**MDLM Evaluation Guidelines.** Our findings motivate two key recommendations for evaluating MDLMs. First, benchmark reports should explicitly state the number of mask tokens used during evaluation. Our work suggests that this detail is critical for reproducibility, yet we observe that it is often omitted in prior work (Ye et al., 2025; Song et al., 2025). Second, we advocate incorporating mask-sensitivity analysis as a standard component of MDLM evaluation pipelines, conducted specifically on tasks requiring long-context comprehension. Popularising such analysis could reveal how novel decoding strategies and post-training methods influence robustness to variations in mask count, fostering a deeper understanding of model behaviour under realistic usage conditions.

**Limitations.** In this work, we have focused on the analysis of open-source MDLMs. However, we note that important details of the pre-training protocol for these models (e.g., exact datasets used) have not been publicly released. Their differing training data and pipelines likely contribute to some of the behaviours we uncovered, and a clearer picture could emerge with access to these details. Understanding how specific data distributions, masking schedules, and architectural choices interact with the diffusion objective would help disentangle model-specific quirks from general properties of MDLMs. This represents a limitation of our study, as a more complete understanding of context processing in diffusion models would benefit from controlled comparisons across models with fully transparent training setups.

**Future Work.** In future work, we would like to examine uniform diffusion models (Lou et al., 2024), which avoid explicit masks and instead apply noise more evenly across the input. If such models prove more robust to context placement and less sensitive to decoding configurations, this could clarify whether the issues we identify are intrinsic to the diffusion paradigm or specific to masked variants. While we document strong locality biases in MDLMs, the mechanisms behind these biases remain unclear. A deeper analysis of the masked diffusion objective, particularly its weighting schemes and noise schedules, could help explain why models favour nearby context and suggest ways to adjust training dynamics for fewer positional biases.

## Acknowledgments

We thank Dana Kianfar and Usman Anwar for their insights offered during the initial stages of this work.

## Impact Statement

This paper presents work whose goal is to advance the field of Machine Learning by identifying and addressing fundamental limitations in Masked Diffusion Language Models (MDLMs). Our findings reveal that MDLMs exhibit locality biases and are sensitive to mask token configurations, which has important implications for their deployment in real-world applications. As with all language model research, our work could be used to improve AI systems that may have both beneficial applications (e.g., more efficient text generation, better context understanding) and potential risks (e.g., if deployed without proper safeguards). We believe the transparency provided by our analysis of model limitations contributes positively to the responsible development of language models. There are no specific ethical concerns beyond those common to language model research in general.

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

# Appendix Contents

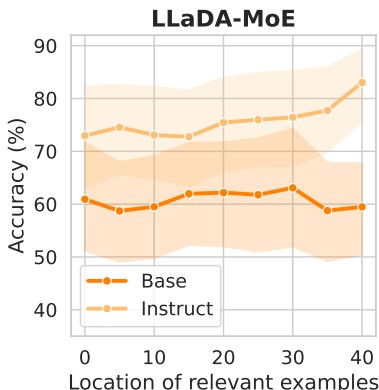

*Figure 10.* **Recency bias in LLaDA-MoE (re: Fig 1).** LLaDA-Moe-Instruct displays a strong recency bias, as seen also in other MDLMs, while the performance of LLaDA-MoE-Base is more agnostic to the location of relevant examples within the context.

## A. Additional Experimental Results

### A.1. Results on LLaDA-MoE

**Motivation.** As the area of MDLMs is still in early stages of development, the number of open-source MDLMs available for evaluation is still heavily limited. In our work, we follow the example of existing works and conduct all the evaluations on the LLaDA and Dream models (Shansan et al., 2025; Israel et al., 2025; Li et al., 2025; Wang et al., 2025). We believe that the identified limitations of these models, particularly given their prevalence, can guide training and deployment of future MDLMs and hence significantly contribute to the field. To improve the generalisability of our results, we have rerun the experiments in sections Section 4 and Section 5 of the paper also on LLaDA-MoE (Zhu et al., 2025)–a mixture of experts MDLM, providing significant training details in the provided model report. Importantly, **LLaDA-MoE has been fine-tuned to context lengths of 8k**, thus increasing the context length compared to LLaDA and Dream.

**Results.** Figures 10-16 show the results of our analysis conducted on LLaDA-MoE. LLaDA-MoE largely displays patterns similar to that of LLaDA, although some results merit further discussion. In Figures 10 and 11 we note that LLaDA-MoE-Base does not display a significant recency bias (its performance is mostly agnostic to the location of relevant examples). We hypothesise that this might be because LLaDA-MoE was fine-tuned to handle context lengths of 8k, which is significantly more than the length of the tasks considered in our evaluation. Nevertheless, the gradient attribution analysis (Figure 19) still demonstrates patterns consistent with the recency bias present in other MDLMs, suggesting that this issue might still affect performance in tasks with longer input. **The fine-tuning experiment with the MA loss (Figure 16) clearly demonstrates that the MA loss can be effective in reducing the negative effect of extra masks**.

**Remark.** We note that the performance of LLaDA-MoE presented in Figure 10 does not align exactly with the performance for the case when we use 2.0 masks in Figure 15. We note that this discrepancy stems from the fact that in Figure 10 we use only a single mask, followed by the end of sentence, rather than two separate masks (see Section C.4 for details). This discrepancy indicates that LLaDA-MoE is highly sensitive to the number of masks, and small variations can significantly affect performance, further reiterating the importance of our findings.

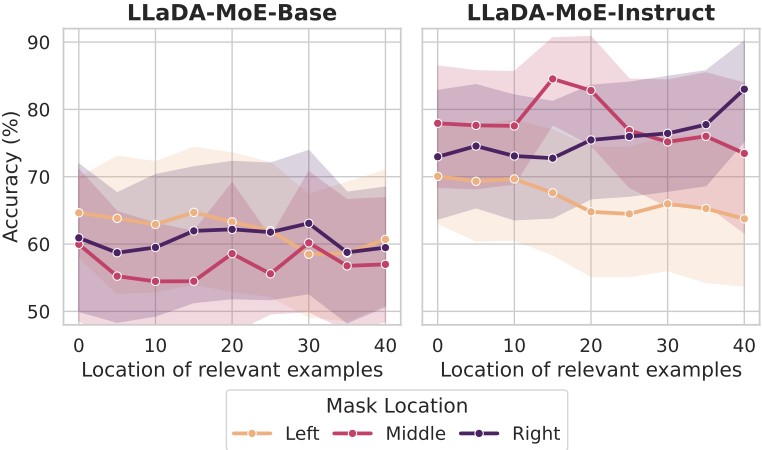

Figure 11. **Locality bias in LLaDA-MoE (re: Fig 2).** LLaDA-MoE-Instruct displays a strong locality bias, as seen also in other MDLMs (the performance is best when the relevant examples are located close to the masked question). The performance of LLaDA-MoE-Base is more uniform across the locations of relevant examples, although at a significantly lower level overall.

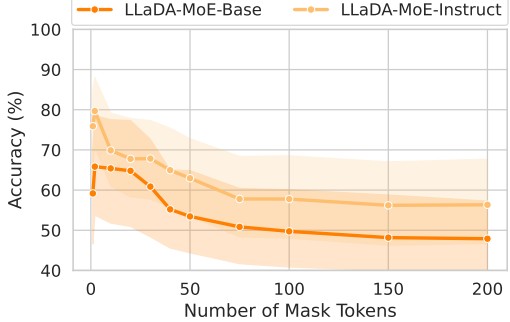

Figure 12. **Performance of LLaDA-MoE decreases significantly with added masks (re: Fig. 4).**

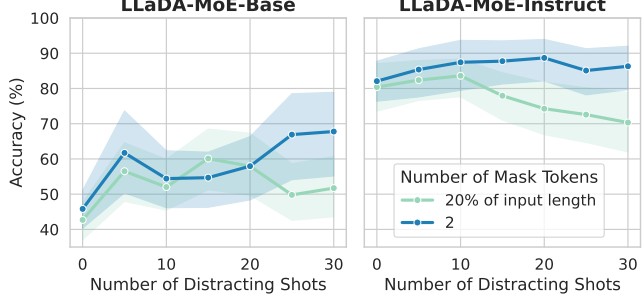

Figure 13. **For LLaDA-MoE, the performance degradation becomes more significant as the context length increases (re: Fig 5).** This effect is particularly visible in LLaDA-MoE-Instruct.

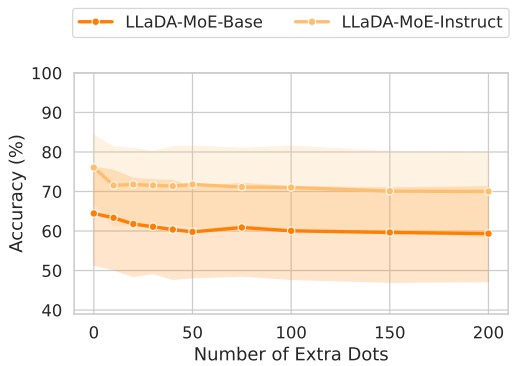

*Figure 14.* **For LLaDA-MoE, extra dots do not degrade performance as strongly as extra masks (re: Fig 6).**

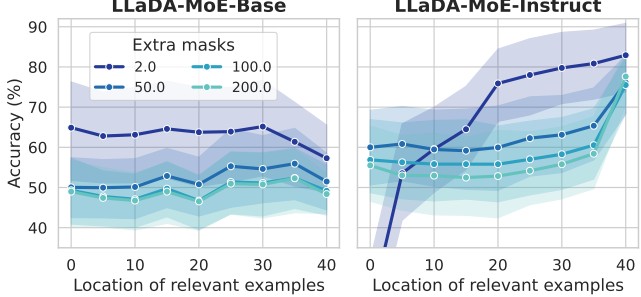

*Figure 15.* **Extra masks alter the locality bias in LLaDA-MoE (re: Fig. 8)**. For both the Base and the Instruct model, the performance becomes significantly worse as we add extra masks, across all locations. For LLaDA-MoE-Instruct in particular, the performance is more uniform across most locations with the extra masks.

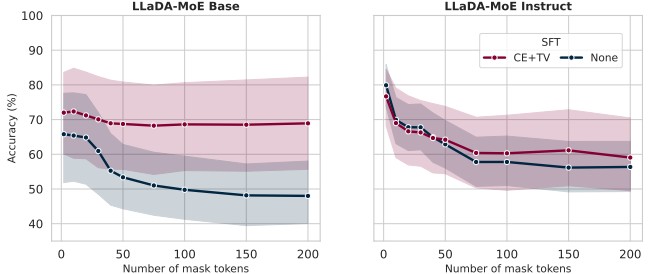

*Figure 16.* **MA loss can rectify the effect of extra masks.** Particularly in the LLaDA-MoE-Base model, fine-tuning with the MA loss allows to induce the robustness to extra masks, leading to improved performance. We use random unmasking strategy.

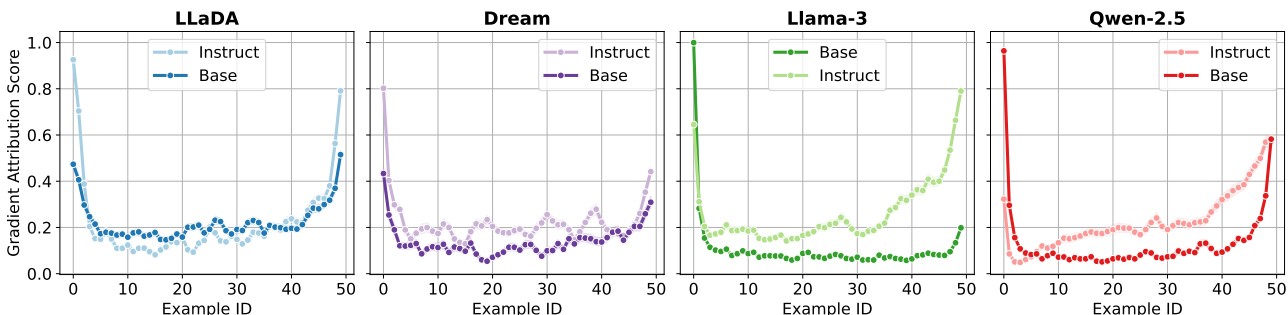

*Figure 17.* **Gradient attribution analysis further illuminates the locality bias of the models.** The figure shows the results of the gradient attribution analysis when the question of interest is located at the **right** end of the context. Although all models display the characteristic U-shaped behaviour, MDLMs demonstrate more uniform gradients across different positions, indicating reduced locality bias compared to their ARLM counterparts.

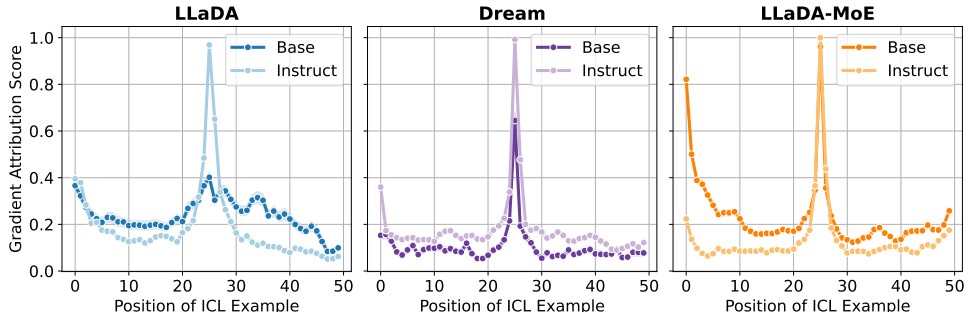

*Figure 18.* **Gradient attribution analysis confirms locality bias in MDLMs.** Normalised gradient attribution results for when the target question is placed in the **middle** of the input context showcase a strong recency bias.

## A.2. Gradient Attribution Analysis of MDLMs

Here we provide more details and additional results for the gradient attribution analysis presented in Sections 4.3 and 5.3.

**Results.** Figures 17 - 20 show *normalised* gradient scores across the different in-context examples (gradient scores after subtracting the minimum value and dividing by the range calculated for each model and each dataset, such that the final values are in the range $[0, 1]$). Figures 20 and 18, where the masked question is located in the middle and to the left respectively, further show a clear locality bias of the studied MDLMs: the normalised gradients have consistently larger values at positions closer to the mask of interest (i.e. for positions 20-30 when the masked question is located in the centre of the input, and for positions 0-10 when the masked question is located on the left end of the input). This provides additional evidence for our results presented in Section 4, indicating that MDLMs display a strong locality bias. Figure 19 also demonstrates similar patterns for LLaDA-MoE. For completeness, Table 2 shows also the attention attributed to mask tokens for LLaDA-MoE (extending the results in Table 1).

| Model Name | Masks | Non-Masks (Last 50) | Non-Masks |
|---|---|---|---|
| Dream-Base-7b | $0.282 \pm 0.040$ | $0.012 \pm 0.007$ | $0.005 \pm 0.003$ |
| Dream-Instruct-7b | $0.144 \pm 0.031$ | $0.030 \pm 0.005$ | $0.018 \pm 0.002$ |
| LLaDA-Base-8b | $0.234 \pm 0.021$ | $0.005 \pm 0.002$ | $0.005 \pm 0.002$ |
| LLaDA-Instruct-8b | $0.220 \pm 0.031$ | $0.057 \pm 0.014$ | $0.017 \pm 0.003$ |
| LLaDA-MoE-Base | $0.237 \pm 0.034$ | $0.094 \pm 0.016$ | $0.029 \pm 0.003$ |
| LLaDA-Moe-Instruct | $0.188 \pm 0.032$ | $0.150 \pm 0.024$ | $0.028 \pm 0.004$ |

*Table 2.* **MDLMs are particularly sensitive to mask tokens.** We show the average normalised gradients attributed to the mask tokens, compared to all the other tokens in the input sequence.

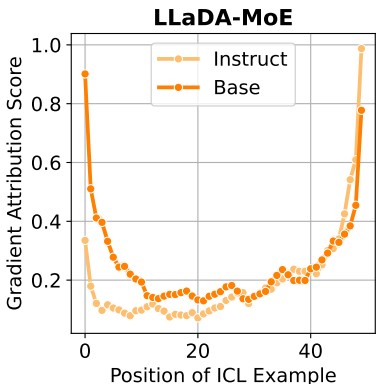

*Figure 19.* **Gradient attribution analysis reveals recency bias in LLaDA-MoE (re: Fig 3).** Both LLaDA-MoE-Base and LLaDA-MoE-Instruct display a strong recency and primacy bias, based on the gradient attribution analysis conducted when the masked question is placed on the *right* end of the context (similarly as in Figure 17).

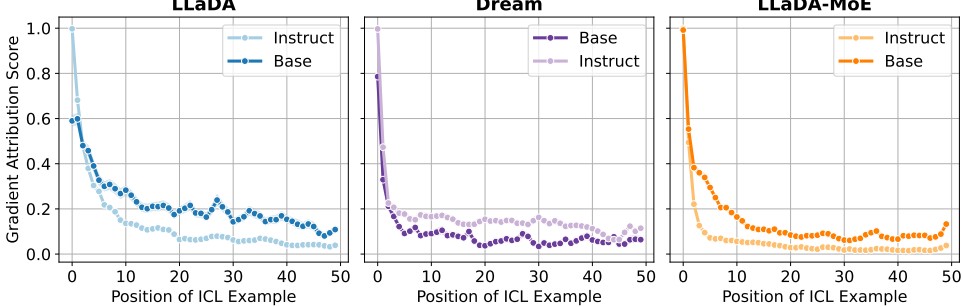

*Figure 20.* **Gradient attribution analysis confirms locality bias in MDLMs.** Normalised gradient attribution results for when the target question is placed on the **left** end of the input context. The results show a strong recency bias, but without the characteristic U-shaped behaviour typically attributed to the primacy bias.

## A.3. Correlation Between Mask Degradation and Context Significance

**Extra Masks Hurt Behaviour On Tasks Requiring Long Context Comprehension.** In Section 5.2, we presented initial evidence that additional masks impair the model's ability to utilise long contexts. Here, We investigate this effect further by analysing the performance of MDLMs on a variety of few-shot learning tasks with single-token answers.

**Setup.** For each task, we evaluate performance along two axes. First, we measure the gain in accuracy when increasing the number of in-context examples from 5 to 25, which serves as a proxy for the task's dependence on long-context information. Second, we compare performance between two masking configurations–one with a single extra mask and one with 200 extra masks–using the 25-shot setting. This quantifies the degradation in predictive accuracy induced by extra masks.

**Results.** Figure 21 visualises the relationship between performance gains from additional in-context examples and degradation due to extra masks (both

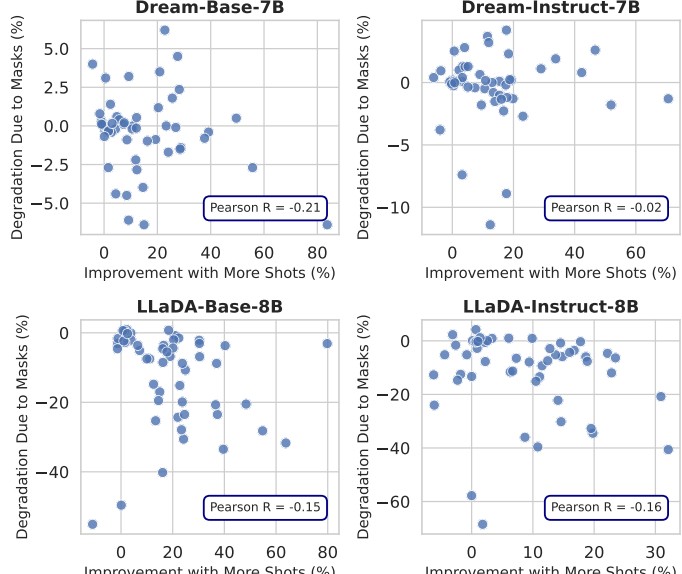

*Figure 21.* **For LLaDA models, tasks that benefit more from additional ICL shots exhibit stronger performance degradation under extra masks.** Dream shows no such trend, remaining more robust to extra masks.

expressed as absolute accuracy differences). For LLaDA-Base and LLaDA-Instruct, most points lie below the $y = 0$ line, indicating substantial degradation—up to 60% on some tasks. The negative Pearson correlations ($R = -0.15$ and $R = -0.16$, respectively) suggest that tasks benefitting most from longer contexts are also those most affected by extra masks. While the correlations are modest, they reinforce the hypothesis that masking disproportionately disrupts long-context processing, though other factors likely also determine the level of degradation.

By contrast, Dream models show minimal and less consistent degradation ($\leq 12\%$), aligning with our earlier observation that MDLMs initialised from autoregressive (AR) weights exhibit increased robustness to masking effects.

**Details of the few-shot learning tasks used.** Each point on the scatterplots presented in Figure 21 corresponds to a different few-shot learning task. We use the following few-shot learning datasets investigated in the different sections of the paper: (1) The pattern recognition tasks described in Section 3 (16 combinations). (2) All the variants of the multi-dimensional classification dataset described in Section A.10. Additionally, we use the following popular ICL datasets: AG News (Zhang et al., 2015), SST-2 (Socher et al., 2013), Rotten Tomatoes (Pang & Lee, 2005), as well as MRPC, RTE and QNLI from GLUE (Wang et al., 2018). For AG News, we restrict the dataset to three categories only (excluding 'Science and Technology') such that each of the correct labels can be expressed with a single token only. For RTE dataset, we use the original validation set for getting the in-context examples and use the train set as the evaluation set, to maximise the number of examples in the evaluation set. For datasets where the examples are ordered by label, we shuffle the datasets upon loading to ensure that there is an even distribution between the different classes within the in-context examples provided to the models. For RTE, QNLI and AG News datasets we also filter the examples in the train and test sets such that the length of the text does not exceed 500 characters.

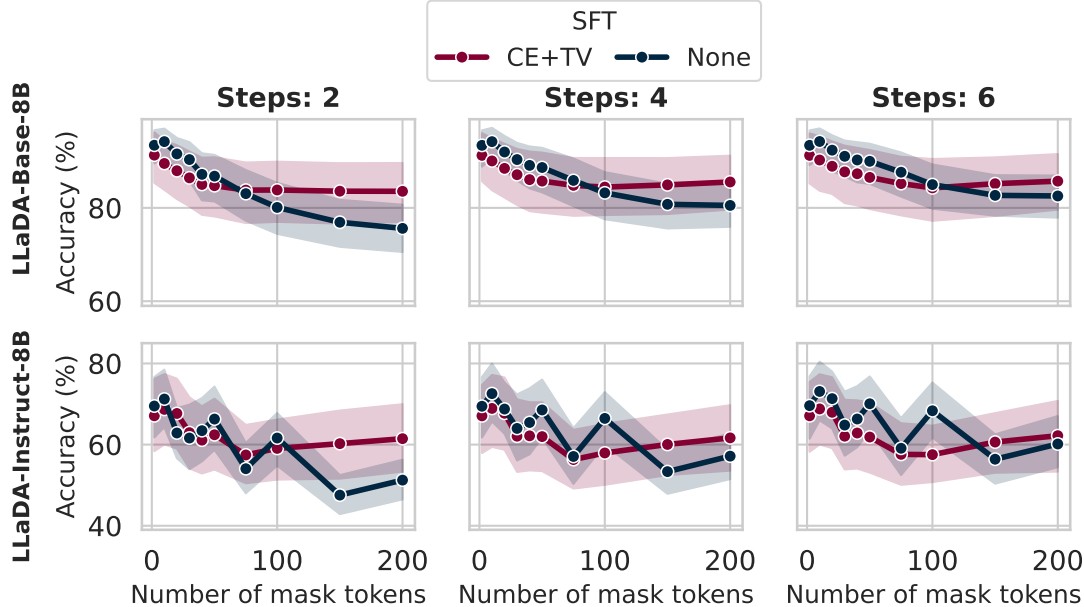

*Figure 22.* Performance across varying numbers of mask tokens for different decoding steps (2, 4, and 6) using random unmasking strategy, for LLaDA models. The base model (None) shows significant performance degradation as the number of mask tokens increases, while our CE+TV fine-tuned model maintains more stable performance across all configurations.

### A.4. Robustness Analysis: Decoding with Few Steps on the LLaDA Models

**Motivation.** Our results in Figure 6 demonstrated that when using 40 decoding steps, the performance degradation due to extra masks is alleviated. In Figure 9 we showcased that the same effect can be achieved in just 1 decoding step with the help of our mask-agnostic fine-tuning, achieving much lower latency. Here, we provide intermediate results showing how performance varies when using 2, 4, and 6 decoding steps to further evaluate the benefits of using the mask-agnostic loss.

**Results.** Figure 22 shows the results obtained using the random unmasking strategy (tokens were unmasked in random order) for LLaDA models, while Figure 23 shows similar results for LLaDA-MoE-Base. Across all configurations, we observe that increasing the number of mask tokens leads to performance degradation in both the base model and our fine-tuned model. However, the CE+TV fine-tuned model consistently maintains higher performance and exhibits significantly less degradation.

**Robustness Analysis.** To quantify this improved robustness, we measure the relative performance degradation as the percentage drop from maximum to minimum accuracy: $\frac{\text{max accuracy} - \text{min accuracy}}{\text{max accuracy}} \times 100\%$. As shown in Figure 24, our CE+TV fine-tuning substantially reduces performance degradation across all step configurations:

- **LLaDA-Base-8B**: Degradation reduced from 15.5% to 7.9% (49% reduction)

- **LLaDA-Instruct-8B**: Degradation reduced from 27.5% to 17.0% (38% reduction)

Importantly, this improved robustness comes with minimal accuracy trade-offs. The CE+TV model maintains competitive or superior performance at low mask token counts while being significantly more robust as the number of mask tokens increases. This demonstrates that our mask-agnostic fine-tuning not only enables efficient single-step decoding but also fundamentally improves the model's ability to handle varying numbers of mask tokens, making it more practical for real-world applications where computational constraints may vary.

However, we emphasise that while our method mitigates the degradation due to extra masks, it does not fully eliminate it. The fact that a non-negligible performance drop persists–even after targeted fine-tuning and multiple decoding steps–underscores the severity of the mask distraction phenomenon. It suggests this is not a trivial artifact, but a deep-seated characteristic of current MDLM architectures that cannot be easily ignored and requires continued investigation.

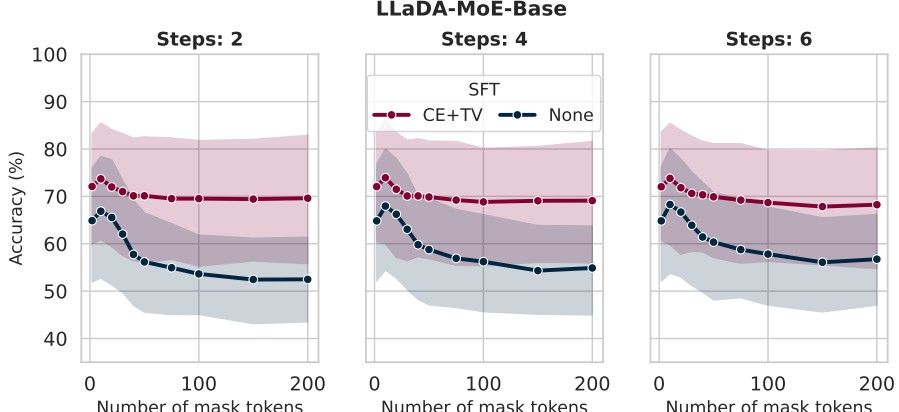

*Figure 23.* Performance across varying numbers of mask tokens for different decoding steps (2, 4, and 6) using random unmasking strategy, for LLaDA-MoE-Base model. The MA loss helps to rectify the negative effect of extra masks across all numbers of decoding steps considered.

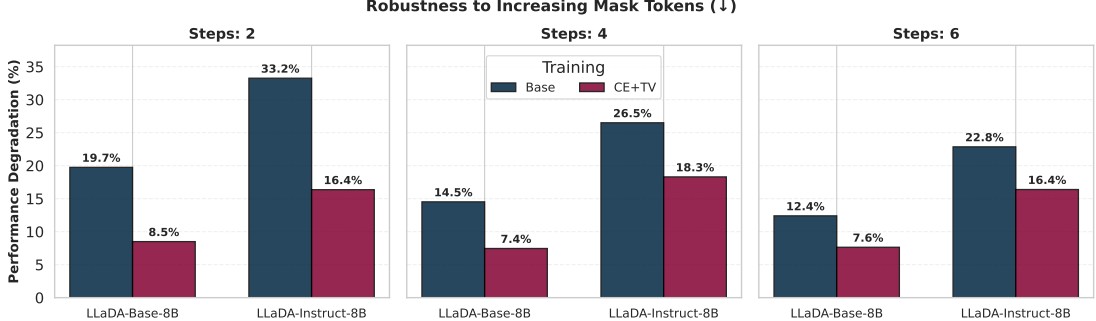

*Figure 24.* Relative performance degradation (measured as $\frac{\text{max accuracy} - \text{min accuracy}}{\text{max accuracy}} \times 100\%$) across different numbers of decoding steps. Lower values indicate better robustness to increasing mask tokens. Our CE+TV fine-tuning reduces degradation by 38-49% compared to the base model, demonstrating significantly improved robustness with minimal accuracy trade-offs.

*Table 3.* **Training using the MA loss does not significantly affect the language modelling performance.** We evaluate the performance of the different LLaDA models before fine tuning with the MA loss (No LoRA) and after fine tuning (LoRA). The results present the average values computed over 250 samples from each dataset, with the standard errors. We observe no significant different in language modelling performance. The generative perplexity (PPL) is computed using Qwen2.5-Base-8B as a reference model.

| Dataset | Model | ROUGE-L (↑) | | | Generative PPL (↓) | | |
| | | No LoRA | LoRA | Δ | No LoRA | LoRA | Δ |
|---|---|---|---|---|---|---|---|
| wikitext | LLaDA-Base-8B | $0.497 \pm 0.010$ | $0.486 \pm 0.010$ | -0.011 | $6.013 \pm 0.120$ | $6.061 \pm 0.110$ | +0.048 |
| wikitext | LLaDA-Instruct-8B | $0.489 \pm 0.010$ | $0.474 \pm 0.009$ | -0.015 | $6.039 \pm 0.108$ | $6.087 \pm 0.109$ | +0.048 |
| wikitext | LLaDA-MoE-Base | $0.502 \pm 0.009$ | $0.497 \pm 0.009$ | -0.005 | $6.466 \pm 0.138$ | $6.191 \pm 0.118$ | -0.276 |
| wikitext | LLaDA-MoE-Instruct | $0.493 \pm 0.009$ | $0.491 \pm 0.009$ | -0.002 | $6.644 \pm 0.131$ | $6.132 \pm 0.115$ | -0.512 |
| lm1b | LLaDA-Base-8B | $0.484 \pm 0.010$ | $0.474 \pm 0.010$ | -0.010 | $6.058 \pm 0.162$ | $6.252 \pm 0.161$ | +0.194 |
| lm1b | LLaDA-Instruct-8B | $0.480 \pm 0.010$ | $0.461 \pm 0.010$ | -0.019 | $6.171 \pm 0.150$ | $6.114 \pm 0.141$ | -0.057 |
| lm1b | LLaDA-MoE-Base | $0.469 \pm 0.010$ | $0.468 \pm 0.010$ | -0.001 | $6.241 \pm 0.165$ | $5.828 \pm 0.140$ | -0.413 |
| lm1b | LLaDA-MoE-Instruct | $0.463 \pm 0.010$ | $0.462 \pm 0.010$ | -0.001 | $6.352 \pm 0.159$ | $5.954 \pm 0.147$ | -0.397 |

## A.5. Robustness Analysis: Language Modelling Performance

**Experimental Setup.** We evaluate the language modelling performance on the Wikitext and LM1B datasets (standard language modelling datasets, that were not used for training using the MA loss). To ensure a fair comparison across all models, we consider a setting aligned with the training objective of MDLMs, namely token infilling under heavy masking. For each text sample, we first sample the masking probability uniformly from the interval [0.6, 0.9], and then independently mask input tokens according to this probability. This results in inputs with 60–90% masked tokens (on average) at random positions, requiring substantial reconstruction of global context. We then perform iterative unmasking using top-1 decoding per each token, and the high-confidence decoding scheme between tokens, unmasking one token at a time. We evaluate on 250 samples per dataset, restricting sequence length to 100–400 tokens to control computational cost. We report two complementary metrics: ROUGE-L (measuring the longest common subsequence between the generated text and a reference) and Generative PPL (calculated as the perplexity of a reference model, in our case Qwen2.5-Base-8B, on the text generated by the model being evaluated). This combination allows us to jointly assess faithfulness (ROUGE-L) and linguistic plausibility (Gen-PPL).

**Results.** The results, presented in this table, demonstrate that the MA loss has little to no significant effect on the language modelling capabilities of the models. These findings suggest that incorporating the MA loss does not adversely affect the language modelling behaviour of MDLMs in high-mask infilling regimes. This is consistent with the formulation of the objective, which retains a cross-entropy component that anchors the model to the underlying token distribution.

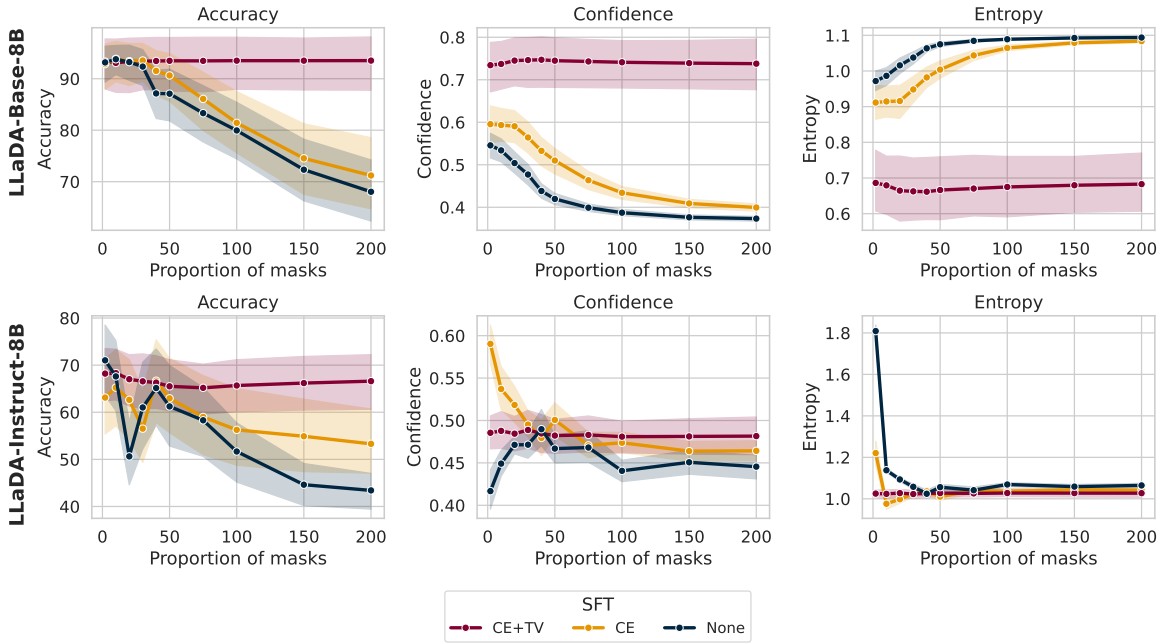

*Figure 26.* **MA loss (CE + TV) decreases the model's entropy and increases the confidence in the generated token**, while making both a smoother function of the number of extra masks, thus increasing the robustness of the model.

### A.6. Additional Results for the Fine-Tuned LLaDA-Instruct

In Figure 25 we provide additional results visualising how the fine-tuning procedure affects the locality of the LLaDA-Instruct model, under different numbers of masks. We observe that the model is more robust to the extra masks, and its performance is more uniform over the different positions of relevant information.

### A.7. Confidence and Entropy as a Function of Masks

In Figure 26 we plot the effect of fine-tuning the LLaDA models with the MA loss on the confidence (calculated as the probability of the generated token, under the greedy decoding scheme) and the entropy of the model's generations. We observe that training with the MA loss significantly increases the confidence in the generated answer for the Base model, and makes the confidence more smooth as a function of extra masks for both models. Furthermore, MA loss also significantly decreases the entropy for both models, also making it more smooth.

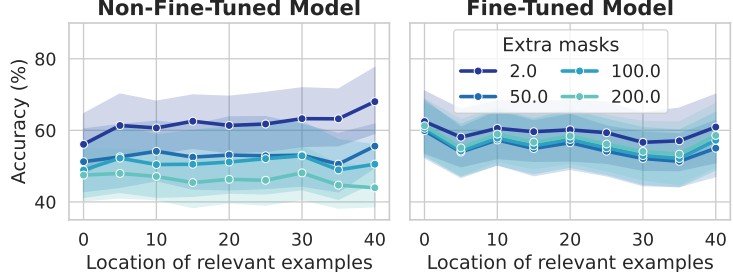

*Figure 25.* **MA loss (CE+TV) reduces the degrading effect of extra masks in LLaDA-Instruct**, and removes the locality of model, however, at the cost of a slight performance decrease.

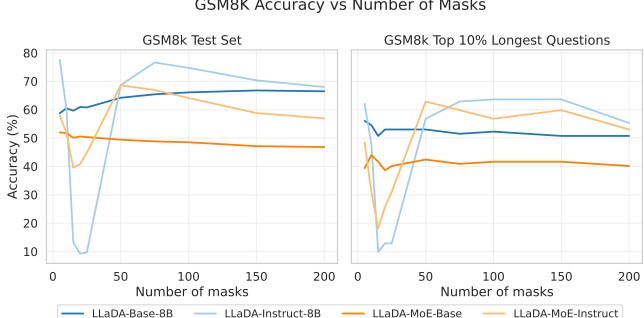

*Figure 27.* **On GSM8k, the performance of LLaDA and LLaDA-MoE varies significantly across different mask configurations**, particularly for the Instruct models. The decreasing performance trend is noticeable in particular when we look at the top 10% of longest GSM8k questions (right panel). This agrees with previous results in Figure 21, which demonstrate that performance degradation is stronger in tasks requiring more context comprehension.

### A.8. Experiments on the GSM8k dataset

**Motivation.**   To further establish whether the lack of robustness to the mask configurations at inference time generalises beyond the in-context learning setting studied, we conduct additional analysis on the GSM8k dataset (Cobbe et al., 2021), consisting of grade-school maths word problems.

**Experimental Procedure and Results.**   To avoid introducing confounding factors, we restrict attention to a 1-step decoding setup. To allow for evaluation in this setting, rather than requiring MDLMs to generate full reasoning chains, we prepend partial solutions (from the original dataset) to each question and task the model with generating only the final answer (which can span no more than 9 tokens). After generation, we proceed to extract the first word from the generated string using regex rules. We evaluate LLaDA and LLaDA-MoE on this task. As visible in Figure 27, the performance of both models—particularly the instruction-tuned variants—is sensitive to the number of masks appended to the input. Further, we evaluate the performance of LLaDA trained with the MA loss on the OpenOrca dataset (the checkpoints analysed in the main paper). We observe that while the performance flattens, it also decreases significantly overall. We hypothesise that this is because fine-tuning on an instruction-following dataset might deteriorate mathematical abilities. To further validate this claim, we then also train the LLaDA models on the GSM8k train dataset directly. As demonstrated in Figure 28, this training indeed effectively makes the models invariant to the mask configurations while increasing overall performance. We further compare to a baseline training run on the GSM8k using the CE loss only and not varying the numbers of masks appended to the input ($l_1 = l_2$). We observe that for LLaDA-Instruct in particular, such training does not allow to reduce sensitivity to masks effectively.

**Experimental Details.**   For training the models on the GSM8k dataset, we follow largely the same setup as for the training on the OpenOrca dataset. The exact task we fine-tune on is predicting the answer, given the input question and the partial answer. For both the base and the instruct models we use the following hyperparameters: $\alpha = 1, \beta = 10, \text{learning rate} = 5 \times 10^{-6}, \text{lower } p = 0.5, \text{upper } p = 0.8, \text{Max Steps Mask Curriculum} = 24, \text{batch size} = 129$. Other hyperparameter choices overlap with those used for the training on OpenOrca (see Table 4). We train for $\approx 90$ gradient descent steps (no more than 2 epochs).

**Discussion.**   The results on GSM8k further corroborate the findings that the masked diffusion models are not always robust to the masking configurations seen during training. However, the observed performance patterns (particularly for the Instruct models) are qualitatively different than those observed for the in-context learning tasks. Unfortunately, the LLaDA and LLaDA-MoE papers and associated codebases provide only limited details about the pre-training and fine-tuning procedures, which makes it difficult to precisely attribute the observed artefacts. In particular, dataset-level information (e.g., distribution of the length of question-answer pairs) that could help explain this behaviour is not available. Despite this limitation, we summarise below several observations and speculative intuitions regarding the possible origins of the observed behaviour on GSM8k.

For both LLaDA and LLaDA-MoE, the SFT stage differs from pre-training in how sequence length is handled. While pre-training operates on fixed-length sequences, SFT uses batch-dependent lengths, where shorter question–answer pairs are

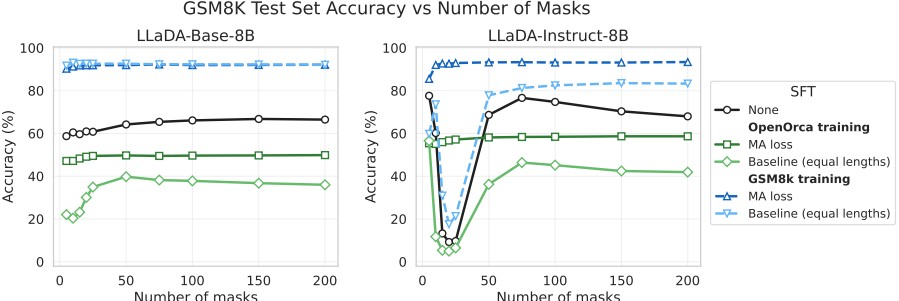

*Figure 28.* **MA loss successfully alleviates sensitivity to mask configurations.** When trained on the GSM8k train set, the MA loss successfully reduces the sensitivity to the number of masks without compromising the overall performance. This gives significant improvement over the baseline particularly for the LLaDA-Instruct model. Training on a general instruction-tuning dataset (OpenOrca) can deteriorate GSM8k performance overall, although even in this case we observe that MA loss reduces sensitivity to masks and improves upon the baseline.

padded with |EOS| tokens to match the longest sequence in the batch.

Due to this preprocessing, the model never observes fewer masks than required to generate the true answer (it sees either the exact number or more, due to |EOS| padding). This may induce spurious correlations between the number of available mask positions and properties of the expected output, without masks carrying meaningful semantic information. For instance, the model may implicitly learn that the correct answer should not exceed the number of available mask positions. Importantly, this would not make masks informative in a useful sense, but could amplify the model's sensitivity to their presence. This effect may be more pronounced in instruct-tuned models, which are known to rely more on surface-level heuristics and formatting cues than base models (Irsoy et al., 2025).

In the context of GSM8k, this bias may manifest as follows: when only a small number of masks is appended to the question and reasoning trace, this aligns with the model's expectation of producing a short answer. Conversely, a large number of masks may signal the need for a longer generation (e.g., an extended reasoning trace). The intermediate regime ($\approx 25\check{}30$ masks) may therefore be ambiguous: too many masks to safely ignore, yet insufficient to clearly trigger longer-form generation. This confusion could explain the observed dip in performance. We emphasise, however, that this remains a speculative hypothesis; access to statistics such as the distribution of sequence lengths in the SFT data would be required to validate it.

Finally, we note that across all masking regimes considered, we were consistently able to extract a valid integer prediction from the decoded outputs. In particular, the models did not degenerate into continuing the chain-of-thought instead of producing a final answer, suggesting that the degradation is due to reasoning quality rather than output formatting.

While this analysis is necessarily speculative, we hope it provides useful intuition about how aspects of the SFT procedure in LLaDA and LLaDA-MoE could give rise to the observed behaviour.

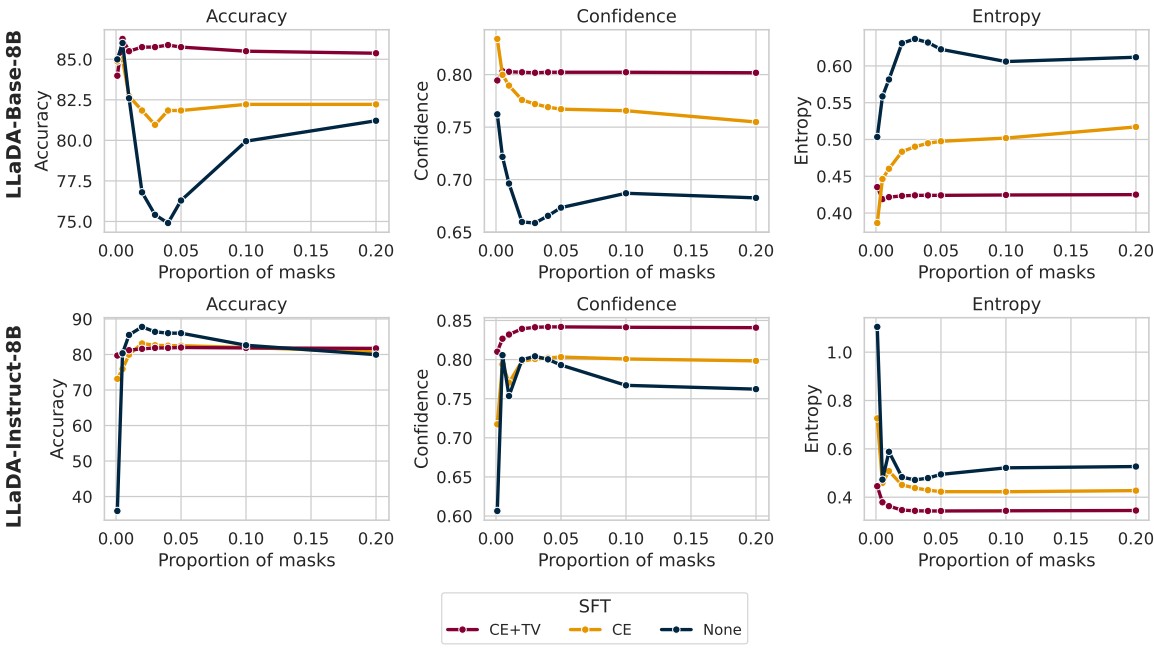

*Figure 29.* **On the HotPotQA dataset, the MA loss also improves the robustness of the models to the varying number of masks.** We observe improved performance particularly for the LLaDA-Base model.

### A.9. Experiments on the HotPotQA dataset

**Motivation.** To further apply whether our results generalise to other in-context learning tasks, beyond the few-shot learning setting, we use a subset of the HotPotQA dataset (Yang et al., 2018). This dataset consists of Wikipedia-based question-answer pairs. The questions require finding and *simultaneously* reasoning over multiple supporting documents (facts), thus ensuring that the dataset requires long-context comprehension.

**Dataset.** We utilised the 'distractor' configuration of HotPotQA and loaded it via the Hugging Face datasets library.[1] Our preprocessing focused on extracting binary-choice questions by filtering for examples containing "or" in the question text. Using regular expression pattern matching, we parsed such questions to extract the question stem and two possible options (A and B). We applied additional filtering to remove examples with input lengths exceeding 1000 tokens (to fit within the context window of the studied MDLMs) and those that could not be reliably converted to multiple-choice format. This approach allowed us to work with a standardised set of binary-choice questions from HotPotQA with single-token answers that were suitable for our controlled experiments and could be reliably evaluated using the accuracy metric. For each example, we concatenated the provided supporting facts (context) together with the question:

```
f"**Context**:\n'{entry['context']}'.\n\n"
    + f"**Question**: '{entry['question']}'"
    + f" [A] {entry['option_A']}\n"
    + f" [B] {entry['option_B']}\n"
    + "**Answer**:[{entry['answer']}]"
```

We use a system prompt ("Which of the following answers is true? Respond with [A] or [B].") and append one in-context learning example to ensure that the model can correctly format its answer.

As the input lengths in this dataset are more variable, rather than adding a pre-determined number of masks as in previous experiments, we add a number of masks proportional to the number of tokens in the input text.

---

[1]https://huggingface.co/datasets/hotpotqa/hotpot_qa

**Results.** We evaluated the performance of LLaDA-Base and LLaDA-Instruct, with and without the mask-agnostic fine-tuning. Without the fine-tuning, we observe a high sensitivity of the Base model to the number of extra masks, with performance decreasing sharply when the number of masks is equal to $\approx 5\%$ of the input length (which corresponds to 90-100 tokens). The fine-tuning allows to effectively remove the variability to the extra masks, once again smoothing out the confidence and entropy curves.

For the Instruct model, we note that even before the fine-tuning the model is more robust to the number of extra masks in this setting. However, the MA loss still allows to smooth out the confidence and the entropy of the model. Further, the MA loss makes the model more robust in the case when the number of available tokens is small (1-2) tokens, in which case the original model fails to provide a coherent answer.

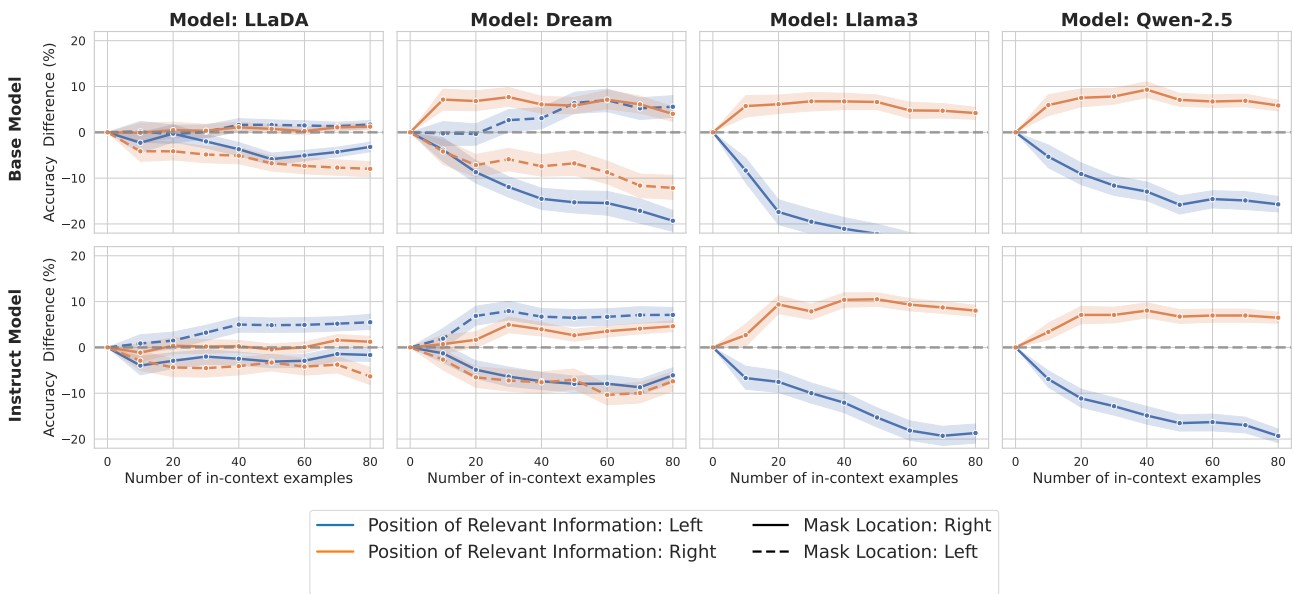

*Figure 30.* **In the multidimensional classification dataset, across all models, performance degrades when the relevant information is distant from the test question.** We report the accuracy difference when placing relevant information on the left versus randomly (blue line), and on the right versus randomly (orange line). For DLMs, we additionally vary the position of the masked question–placing it at either the left or right end of the in-context examples (solid vs. dashed lines). Across all models performance consistently drops when the relevant information is far from the masked question (blue solid and orange dashed lines), with the effect being most pronounced in ARLMs. Notably, Dream exhibits a stronger recency bias when the masked question is positioned on the right than on the left, suggesting an underlying AR bias. *Shaded regions indicate 95% confidence intervals computed across the 4 dataset types, and 5 seeds.*

## A.10. Experiments on the Multi-dimensional Classification Dataset

**Motivation.** While in the pattern recognition tasks presented in the main paper it is relatively clear which examples carry signal for the test question (number vs word tasks), we also consider the setting where this distinction is more blurry, and the contribution of each example to the answer is more ambiguous. Specifically, we construct a multidimensional classification task, where each point is described using a three-dimensional integer coordinates and a binary label. To make the tasks difficult, we use different non-linear decision boundaries, described below. The task of the model is to predict the label for a new point. To measure the sensitivity to the position of information, we manipulate the order in which the points are presented: ordering them either randomly, or by the L2 distance in the input space to the test point.

**Dataset.** To evaluate recency bias, we constructed several synthetic binary classification datasets with varying complexity. Each dataset was designed to present different learning challenges, from nonlinear decision boundaries to complex manifold structures. For reproducibility, we generated each dataset type with 5 different random seeds. We utilised four distinct dataset types in our experiments:

1. **Nonlinear dataset:** This dataset features nonlinear decision boundaries created through polynomial feature transformations. We first generated base features as random integers between 1 and 100. We then augmented these with squared terms and interaction terms between features, creating a nonlinear feature space. The final binary labels were determined by applying a logistic function to a weighted sum of these features (with randomly generated coefficients), followed by thresholding at 0.5.

2. **Swiss-roll dataset:** We employed scikit-learn's `make_swiss_roll` function to generate data points along a 3D swiss roll manifold. The continuous position along the roll (colour parameter) was converted to binary labels by thresholding at the median value, creating two interleaved classes that cannot be separated by a linear boundary. The 3D coordinates were then scaled to integers between 1 and 100 to maintain consistency with our other datasets.

3. **Moons dataset:** Using scikit-learn's `make_moons` function, we created two interleaving half-moon shapes in 2D space. This dataset presents a clear nonlinear boundary challenge. The resulting coordinates were scaled to integers between 1 and 100.

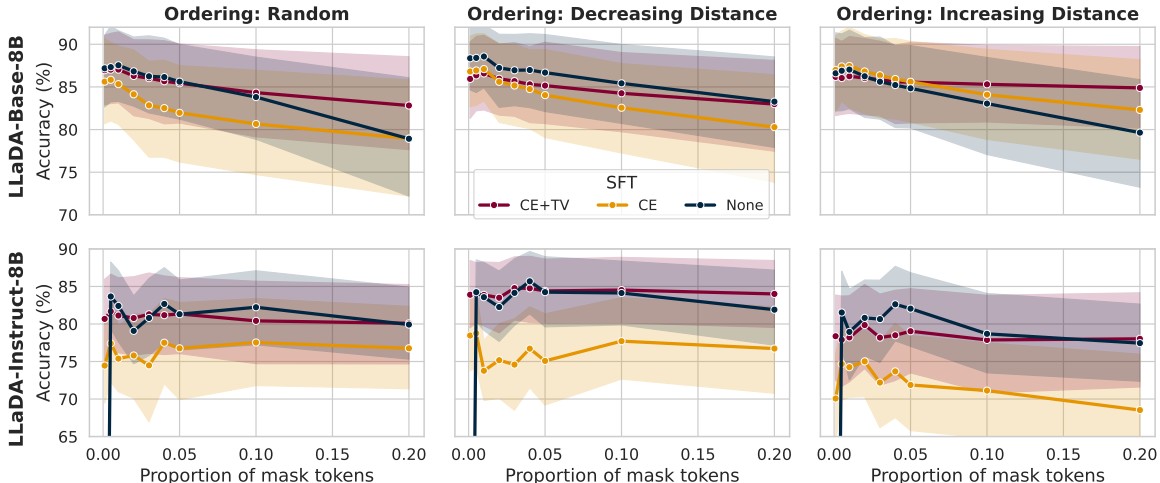

*Figure 31.* **In the multidimensional-classification dataset, the MA loss prevents performance degradation with the extra masks (particularly for the Base model).** We observe how the performance of the models changes under different ordering schemes of the in-context examples: random ordering, ordering by decreasing distance (most relevant information is located close to the test question) and ordering by increasing distance (most relevant information is located far from the test question). *Shaded regions indicate 95% confidence intervals computed across the 4 dataset types, and 5 seeds.*

4. **Circles dataset:** We generated concentric circles using scikit-learn's `make_circles` function, creating another challenging nonlinear classification problem. As with the other datasets, the coordinates were scaled to integers between 1 and 100.

To ensure class balance, we generate equal numbers of positive and negative examples for both training (100 examples) and test splits (1000 examples) of each dataset. Additional dimensions beyond those generated by the base algorithms were filled with random integers, such that each dataset has is three-dimensional. Each input vector is stored as a space-separated string of integers. We use the class labels 'Above' and 'Below'.

**Setup.** To study the recency bias (i.e., whether or not the models have a tendency to pay more attention towards examples which are closer to the generation point), we employ the following ordering schemes to the selected in-context examples:

- **Random ordering:** The in-context examples are ordered randomly.

- **Ordered by distance to the test point, in decreasing order:** When formatting each prompt, we compute the L2 distance of each in-context example to the test example. We then order the in-context examples in decreasing order, such that examples on the far left of the prompt are furthest away from the test point, and examples on the far right are closest in distance to the test point. This corresponds to the 'Position of relevant information: Right' setting.

- **Ordered by distance to the test point, in decreasing order:** We again compute the L2 distance of each in-context example to the test example, but now order the points in increasing order, such that examples on the far left of the prompt are closest to the test point, and examples on the far right are furthest away in distance to the test point. This corresponds to the 'Position of relevant information: Left' setting.

We note that in all settings, the selected in-context examples are fixed, we just change their order within the prompt. Under this setting, a conventional supervised learning algorithm should be agnostic to the ordering of the provided information. We run the experiments with the masked example placed both on the left-end of the prompt and on the right-end of the prompt.

**Results.** Firstly, we use the created setup to further evaluate the locality bias of the MDLMs and ARLMs. Results in Figure 30 show that performance of MDLMs (particularly Dream, initialised with the weights of an ARLM) drops significantly when relevant examples are far from the masked question, confirming a locality bias – though weaker than in ARLMs.

Secondly, we evaluate the robustness of the LLaDA-Base and LLaDA-Instruct models to the varying numbers of masks under the different ordering schemes. We focus on the case with 30 in-context examples, with the test question located on the right-end of the in-context examples. Results in Figure 31 show that for the Base model, our MA loss prevents performance degradation, particularly for the random ordering and the ordering by increasing distance (where the most relevant information is far away from the test question). For the Instruct model, our fine-tuning scheme improves robustness to the number of masks, and consistently prevents significant performance degradation with small numbers of masks.

# B. Details of the Supervised Fine-Tuning Pipeline

Below, we present the pseudo-code for calculating our MA loss, and list the hyperparameters we used during fine-tuning. We use batch-size of three, and we pad the inputs with the end of sequence tokens to ensure equal lengths of the input. Additionally, to make training more stable, we introduce a curriculum for the lengths of the masks added at the end of the inputs, starting from minimal numbers of extra masks, and reaching 600 masks over 5000 gradient descent steps. As in our language modelling setup $p_\theta$ is a categorical distribution, we compute the TV distance $TV\left(p_\theta(x^j|x_1), p_\theta(x^j|x_2)\right)$ as the L1 distance between the probabilities (after softmax) obtained for the two inputs. We conduct the LoRA-based fine-tuning (and the subsequent evaluations of the fine-tuned models) on the non-quantised version of the LLaDA models, to ensure more stable training.

We use the following specific values of the hyperparameters for individual settings, chosen based on the lowest value of the loss functions achieved across the considered settings:

- **Base model, CE loss:** $\beta = 0.0, LR = 10^{-6}$

- **Base model, CE + TV loss:** $\beta = 1.0, LR = 10^{-5}$

- **Instruct model, CE loss:** $\beta = 0.0, LR = 10^{-5}$

- **Instruct model, CE + TV loss:** $\beta = 100.0, LR = 5 \times 10^{-7}$

---

**Algorithm 1** Mask-agnostic training

---

**Require:** $\mathcal{P}$: set of input pairs $(q, a)$
**Require:** $p_l, p_u$: lower and upper probabilities of masking
**Require:** $N$: maximum length of text allowed for the model
**Require:** max_masks: Maximal number of masks to be appended to the input
**Require:** $\alpha, \beta$: regularisation coefficients
  **for** $(q, a)$ in $\mathcal{P}$ **do**
    Sample $p \sim U(p_l, p_u)$.
    Create a noised version of the answer $\tilde{a}$ with masking probability $p$.
    Sample $l_1, l_2 \sim \mathcal{U}\left(0, \min(L - \text{len}(p \oplus a), \text{max\_masks})\right)$.
    $x_1 \leftarrow p \oplus \tilde{a} \oplus ([\text{MASK}] * l_1)$
    $x_2 \leftarrow p \oplus \tilde{a} \oplus ([\text{MASK}] * l_2)$
    Pad $x_1, x_2$ with EOS tokens such that they have equal length.
    $o_1 \leftarrow \text{MDLM}(x_1)$
    $o_2 \leftarrow \text{MDLM}(x_2)$
    Compute the MA loss: $\alpha\mathcal{L}_{TV} + \beta\mathcal{L}_{CE}$.
    **Backpropagate**(Loss).

---

*Table 4.* Fine-tuning hyperparameters

| Category | Parameter | Description | Value |
|---|---|---|---|
| General | Max Context Length | Maximum number of tokens processed in a single forward pass | 1024 |
| | Lower p | Lower threshold for masking probability | 0.2 |
| | Upper p | Upper threshold for probability in sampling | 0.8 |
| Loss | $\alpha$ | Weight coefficient for the CE loss | 0.1 |
| | $\beta$ | Weight coefficient for the TV loss | See B |
| | Max Steps Mask Curriculum | Number of steps for mask curriculum learning | 120 |
| | Max Masks | Maximum number of mask tokens that can appended to the sequence | 600 |
| Training | Batch size | Size of each batch | 258 |
| | Mixed Precision | Numerical precision format used during training | bf16 |
| | Max Gradient Norm | Maximum L2 norm of gradients for clipping | 1.0 |
| LoRA | Rank | Dimension of low-rank adaptation matrices | 64 |
| | $\alpha_l$ | Scaling factor for LoRA adaptation | 128 |
| | Dropout | Probability of dropping neurons during training | 0.0 |
| | Learning Rate (LR) | Step size for optimizer updates | See B |
| | Weight Decay | L2 regularization coefficient | 0.0 |

## C. Experimental Details

### C.1. Models

Throughout our experiments we use the following open-source model families, all accessed via the Huggingface API:

- **LLaDA (Nie et al., 2025):** An 8B diffusion language model pre-trained from scratch using the masked diffusion loss (Sahoo et al., 2024).

- **LLaDA-MoE (Zhu et al., 2025):** A 7B mixture of experts diffusion language model pre-trained from scratch using the masked diffusion loss.

- **Dream (HKU NLP Group):** A 7B diffusion language model, whose weights are initialised from those of an autoregressive Qwen-2.5-7B.

- **Qwen-2.5-7B (Yang et al., 2024; Team, 2024):** A fully AR model.

- **Llama3-8B (AI@Meta, 2024):** A fully AR model, with the architecture similar to that of LLaDA (Nie et al., 2025).

For all models, we use greedy decoding strategy (no sampling). We design all of our experiments in a way such that the correct answer consists of only a single token across all the different models and tokenisers. This is to ensure that our

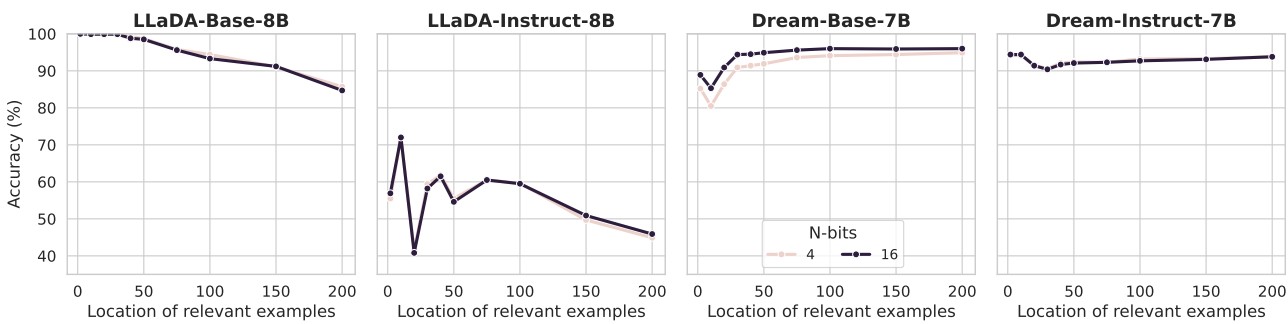

*Figure 32.* **Quantisation has no significant effect on the performance under varying numbers of mask tokens.**

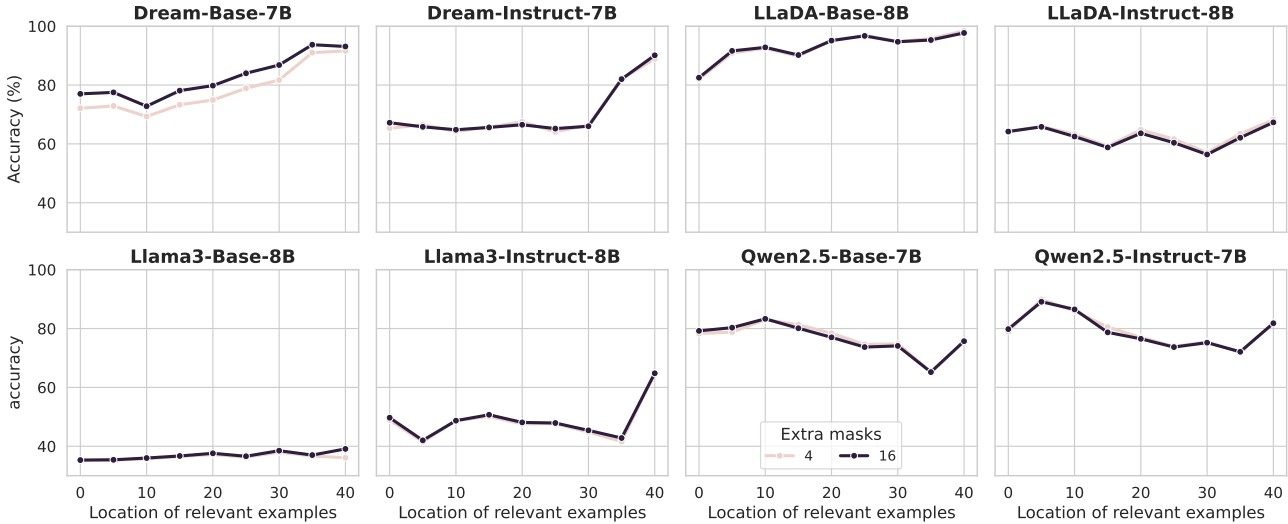

*Figure 33.* **Quantisation has no significant effect on the the locality of the models.**

experiments can isolate the context-processing abilities of the different models, without being confounded by the effect of tokenisation and/or decoding schemes. This is particularly relevant for DLMs, for which the number of masks added to the prompt can constitute a strong prior about the answer.

## C.2. Quantisation

In the experiments which *did not* involve SFT (for which we opted to use the full models), to ensure computational efficiency, we quantised all models to 4-bit precision using the Quanto library. In Figure 32 and Figure 33 we compare the performance of the quantised and non-quantised models on a single task from the pattern recognition suite, verifying that the quantisation has no significant effect on the models' locality bias, nor on the performance degradation under extra masks.

## C.3. Details of the Few-Shot Learning Dataset

Below, we provide further explanations regarding the generation of the few-shot learning tasks used in the main part of the paper. For the relevant (word) tasks, we generate a list of words spanning different categories. We then create the following 8 relevant tasks, by juxtaposing the words from the target category (e.g. adjective) with words from other categories (e.g. verb):

- choose country (out of countries and names),

- choose country (out of countries and names),

- choose capitalised word (out of capitalised and non-capitalised words),

- choose verb (out of verbs, adjectives, prepositions and objects),

- choose adjective (out of adjectives, verbs, prepositions and objects),

- choose animal (out of animals, objects, fruits and sports),

- choose colour (out of colours, animals and objects),

- choose emotion (out of emotions, colours, objects and animals),

- choose object (out of objects, emotions, colours and adjectives).

Additionally, we consider the following distractor (number tasks), where the candidate numbers are integers sampled without replacement from the range 1 to 1000:

- choose smallest number,

- choose largest number.

Each task contains three possible answers (A, B, C) formatted in a way presented in Section 3. To provide further illustration of the dataset considered, below we include an example of the input obtained for a dataset with task "choose verb" and distractor task "choose smallest number", in the settings when the relevant and distractor tasks are mixed (as in Figure 3) or not (as in Figure 1).

### C.4. Formatting of the In-Context Learning Examples

Throughout each experiment, we pre-select a certain group of examples from the specified train set to serve as the in-context learning examples for all the test examples (that is, each test example sees exactly the same in-context learning examples, put in the same order). We always embed the final answer within the square brackets to avoid issues around tokenisation of spaces. For instruct models, each in-context example is formatted as a pair of messages: a user message containing the question and an assistant message containing the answer. The test question is added as the final user message, with the answer prefix included in the assistant's response.

**Autoregressive models.** For ARLMs, we add the full test question and the beginning of the answer (e.g., `"Label:["`) to the final formatted prompt and ask the model to continue the generation. We always decode only one new token.

**Diffusion models.**

- In section 4: to allow to robustly compare performance between different locations of the masked question within the provided in-context examples, we structure the answer of the masked question as `"Answer:[<|mask|>]."` and add this to the prompt, where `<|mask|>` is the textual representation of the mask token, specific to each MDLM. We add exactly one copy of the mask token in between the square brackets. We also use this setup in the experiments with extra dots, appending the dots *after* the closing the bracket of the answer.

- In section 5 and 6 (as well as in other experiments with varying number of extra masks), we use a generation style more resembling the setup for ARLMs, where we add the full test question and the beginning of the answer (e.g., `"Answer:["`) to the final formatted prompt, followed by the specified number of masks. As the closing bracket `].` is typically tokenised as a single token, using this setup with exactly two extra masks allows us to mostly recover the performance seen in the previous setup.

**Extracting answers.** To evaluate the models' accuracy, we perform string matching on the greedily decoded answer (that is, we perform evaluation on the decoded answer, rather than on the generated tokens).

**Changing the location of the masked question.** In Figure 2, to evaluate the sensitivity of the DLMs to the positioning of the mask, we design experiments in which the masked question is placed at different positions within the in-context examples (at the beginning (left) or end (right)).

**Dataset formatting example (mixed tasks)**

```
Options:  (A) 915, (B) 491, (C) 266        Answer:[B].
Answer:[C].                                 Options:  (A) 214, (B) 506, (C) 469
Options:  (A) 610, (B) 222, (C) 307        Answer:[A].
Answer:[B].                                 Options:  (A) 242, (B) 138, (C) 689
Options:  (A) 576, (B) 510, (C) 31         Answer:[B].
Answer:[C].                                 Options:  (A) 159, (B) 51, (C) 824
Options:  (A) 463, (B) 142, (C) 797        Answer:[B].
Answer:[B].                                 Options:  (A) 436, (B) 773, (C) 587
Options:  (A) arrive, (B) thoughtful, (C) near   Answer:[A].
Answer:[A].                                 Options:  (A) 95, (B) 312, (C) 390
Options:  (A) 941, (B) 371, (C) 341        Answer:[A].
Answer:[C].                                 Options:  (A) 30, (B) 982, (C) 727
Options:  (A) 694, (B) 772, (C) 727        Answer:[A].
Answer:[A].                                 Options:  (A) 323, (B) 590, (C) 480
Options:  (A) tall, (B) compete, (C) silly  Answer:[A].
Answer:[B].                                 Options:  (A) 640, (B) 621, (C) 525
Options:  (A) 809, (B) 293, (C) 663        Answer:[C].
Answer:[B].                                 Options:  (A) 464, (B) 836, (C) 125
Options:  (A) 755, (B) 63, (C) 166         Answer:[C].
Answer:[B].                                 Options:  (A) 759, (B) 278, (C) 491
Options:  (A) 450, (B) 398, (C) 750        Answer:[B].
Answer:[B].                                 Options:  (A) 70, (B) 435, (C) 386
Options:  (A) 541, (B) 698, (C) 124        Answer:[A].
Answer:[C].                                 Options:  (A) jar, (B) kiss, (C) thoughtful
Options:  (A) 289, (B) 567, (C) 774        Answer:[B].
Answer:[A].                                 Options:  (A) 733, (B) 603, (C) 211
Options:  (A) reliable, (B) search, (C) zucchini  Answer:[C].
Answer:[B].                                 Options:  (A) 73, (B) 48, (C) 876
Options:  (A) 289, (B) 373, (C) 197        Answer:[B].
Answer:[C].                                 Options:  (A) passionate, (B) lettuce, (C) master
Options:  (A) 402, (B) 785, (C) 467        Answer:[C].
Answer:[A].                                 Options:  (A) 169, (B) 784, (C) 919
Options:  (A) 555, (B) 287, (C) 607        Answer:[A].
Answer:[B].                                 Options:  (A) lucky, (B) train, (C) igloo
Options:  (A) 302, (B) 102, (C) 265        Answer:[B].
Answer:[B].                                 Options:  (A) for, (B) calculate, (C) cube
Options:  (A) 790, (B) 409, (C) 904        Answer:[B].
Answer:[B].                                 Options:  (A) 861, (B) 579, (C) 735
Options:  (A) deliver, (B) graceful, (C) sensitive  Answer:[B].
Answer:[A].                                 Options:  (A) 844, (B) 207, (C) 774
Options:  (A) 143, (B) 388, (C) 159        Answer:[B].
Answer:[A].                                 Options:  (A) 502, (B) 361, (C) 954
Options:  (A) 52, (B) 285, (C) 847         Answer:[B].
Answer:[A].                                 Options:  (A) innocent, (B) relax, (C) upbeat
Options:  (A) 688, (B) 588, (C) 426        Answer:[B].
Answer:[C].                                 Options:  (A) underneath, (B) kill, (C) spicy
Options:  (A) 752, (B) 680, (C) 295        Answer:[B].
Answer:[C].                                 Options:  (A) 935, (B) 501, (C) 459
Options:  (A) 24, (B) 868, (C) 400         Answer:[C].
Answer:[A].                                 Options:  (A) concerning, (B) hate, (C) painting
Options:  (A) 865, (B) 455, (C) 497        Answer:[
```

*Figure 34.* Example input for the few-shot learning tasks, showing the relevant task ("choose verb") mixed with distractor examples ("choose smallest number").

**Dataset formatting example (separated tasks, relevant task in position 0)**

```
Options:  (A) deliver, (B) graceful, (C) sensitive       Answer:[C].
Answer:[A].                                               Options:  (A) 694, (B) 772, (C) 727
Options:  (A) innocent, (B) relax, (C) upbeat            Answer:[A].
Answer:[B].                                               Options:  (A) 733, (B) 603, (C) 211
Options:  (A) jar, (B) kiss, (C) thoughtful             Answer:[C].
Answer:[B].                                               Options:  (A) 214, (B) 506, (C) 469
Options:  (A) reliable, (B) search, (C) zucchini        Answer:[A].
Answer:[B].                                               Options:  (A) 809, (B) 293, (C) 663
Options:  (A) arrive, (B) thoughtful, (C) near          Answer:[B].
Answer:[A].                                               Options:  (A) 865, (B) 455, (C) 497
Options:  (A) passionate, (B) lettuce, (C) master       Answer:[B].
Answer:[C].                                               Options:  (A) 450, (B) 398, (C) 750
Options:  (A) lucky, (B) train, (C) igloo               Answer:[B].
Answer:[B].                                               Options:  (A) 323, (B) 590, (C) 480
Options:  (A) underneath, (B) kill, (C) spicy           Answer:[A].
Answer:[B].                                               Options:  (A) 688, (B) 588, (C) 426
Options:  (A) for, (B) calculate, (C) cube              Answer:[C].
Answer:[B].                                               Options:  (A) 169, (B) 784, (C) 919
Options:  (A) tall, (B) compete, (C) silly              Answer:[A].
Answer:[B].                                               Options:  (A) 790, (B) 409, (C) 904
Options:  (A) 555, (B) 287, (C) 607                     Answer:[B].
Answer:[B].                                               Options:  (A) 30, (B) 982, (C) 727
Options:  (A) 463, (B) 142, (C) 797                     Answer:[A].
Answer:[B].                                               Options:  (A) 73, (B) 48, (C) 876
Options:  (A) 289, (B) 567, (C) 774                     Answer:[B].
Answer:[A].                                               Options:  (A) 402, (B) 785, (C) 467
Options:  (A) 464, (B) 836, (C) 125                     Answer:[A].
Answer:[C].                                               Options:  (A) 289, (B) 373, (C) 197
Options:  (A) 861, (B) 579, (C) 735                     Answer:[C].
Answer:[B].                                               Options:  (A) 935, (B) 501, (C) 459
Options:  (A) 844, (B) 207, (C) 774                     Answer:[C].
Answer:[B].                                               Options:  (A) 24, (B) 868, (C) 400
Options:  (A) 755, (B) 63, (C) 166                      Answer:[A].
Answer:[B].                                               Options:  (A) 436, (B) 773, (C) 587
Options:  (A) 502, (B) 361, (C) 954                     Answer:[A].
Answer:[B].                                               Options:  (A) 143, (B) 388, (C) 159
Options:  (A) 52, (B) 285, (C) 847                      Answer:[A].
Answer:[A].                                               Options:  (A) 640, (B) 621, (C) 525
Options:  (A) 576, (B) 510, (C) 31                      Answer:[C].
Answer:[C].                                               Options:  (A) 941, (B) 371, (C) 341
Options:  (A) 242, (B) 138, (C) 689                     Answer:[C].
Answer:[B].                                               Options:  (A) 95, (B) 312, (C) 390
Options:  (A) 541, (B) 698, (C) 124                     Answer:[A].
Answer:[C].                                               Options:  (A) 70, (B) 435, (C) 386
Options:  (A) 159, (B) 51, (C) 824                      Answer:[A].
Answer:[B].                                               Options:  (A) 752, (B) 680, (C) 295
Options:  (A) 610, (B) 222, (C) 307                     Answer:[C].
Answer:[B].                                               Options:  (A) 759, (B) 278, (C) 491
Options:  (A) 302, (B) 102, (C) 265                     Answer:[B].
Answer:[B].                                               Options:  (A) concerning, (B) hate, (C) painting
Options:  (A) 915, (B) 491, (C) 266                     Answer:[
```

*Figure 35.* Example input for the few-shot learning tasks, showing the relevant task ("choose verb") separated from the distractor examples ("choose smallest number"), with the relevant task in position 0.

