# OpenReview forum: "Masks Can Be Distracting: On Context Comprehension in Diffusion Language Models"
_ICML.cc/2026/Conference — ICML 2026 regular_

### Official Review · Reviewer_LHPR · 2026-02-20

**Soundness:** 4
**Presentation:** 3
**Significance:** 3
**Originality:** 4
**Overall Recommendation:** 5
**Confidence:** 4

**Summary:**

This work examines whether masked diffusion language models (MDLM) exhibit locality biases akin to those seen in autoregressive models, and what role, if any, masked tokens play in MDLMs’ ability to effectively attend to their contexts. Through careful experimental design they find that MDLMs indeed exhibit a locality bias and that surprisingly increasing the number masked tokens adversely affects performance. The authors also propose a “mask agnostic” fine-tuning paradigm and show that it alleviates this adverse effect to some extent.

**Compliance With Llm Reviewing Policy:**

Affirmed.

**Final Justification:**

This paper is well executed and presents important results for the community. During the rebuttal, the authors answered my outstanding questions. I therefore maintain my support for this work.

**Key Questions For Authors:**

- Can the authors explain why LLaDA and Dream base models outperform their instruct counterparts?
- Is it possible that the results in Section 5 are an artifact of the synthetic task? For example, would we see a similar performance degradation if we take the GSM8K test set, for example, and provide the model several-shots and “most of the answer” (i.e., the ground truth reasoning trace, up until the final `\boxed{[MASK]`, for example) and then simply tack on many `MASK` tokens at the end? My sense is that we would not see this similar degradation in the real world setting, so I am curious to see it rigorously tested and to see if it corroborates the authors’ claims or not.

**Limitations:**

Yes

**Strengths And Weaknesses:**

### **Strengths**
- The experimental design is comprehensive and the results are well organized.
- Each section is self-contained with a clear hypothesis, experimental setup, and takeaway.
- The mask-agnostic fine-tuning is well motivated and demonstrates clear improvements where off-the-shelf MDLMs fail.

### **Weaknesses**
- I think that including a Listing in the main paper that more clearly illustrates the inputs for sections 4 and 5 would be highly valuable. Listings 1 and 2 in the Appendix are useful, but quite difficult to parse given that line breaks are depicted in unicode `\n` and not typeset. If the authors could find a compact way to represent how the tasks are presented for MDLMs and how masks are appended for the different settings (section 5) I think that would be very useful towards improving readability.
- The performance results and gradient attribution lend evidence to each empirical claim, however, wouldn’t a more ‘standard’ approach of attention mask analysis be highly relevant here as well?
- Certain terms, especially in the abstract and introduction are not well defined, or somewhat ambiguous. For example (line 16 in abstract) “uniform context utilisation” and (line 27 in abstract) “context comprehension” should be qualified / explained more precisely.
- Additionally, the term “distractor” seems to be somewhat overloaded, referring to both the “needle in a haystack”-type effect of burying the relevant task among irrelevant ones and to the effect of additional masks.
- Figure 1 is used to argue that MDLMs demonstrate a strong locality bias. However, for LLada, for example, the performance curve seems pretty flat for most positions of the relevant examples, with meaningful performance degradation / improvement only observed at the extremes.
- The hypothesis proposed in lines 201-210 (LHS column) could be easily tested, in my opinion. At a small scale two MDLMs could be trained: one with the $1/p$ loss scaling and one without. This would strengthen the author's claim here.

**Minor suggestions / typos:**

- Listings 1 and 2 were quite difficult to read. I would recommend a different layout / presentation format, or potentially these listings can be kept as is, but a trimmed down version that is more clearly formatted can be included for illustrative purposes.
- Throughout, when citing Shi et al 2024 and Sahoo et al 2024, I would recommend to include the concurrent work of Ou et al 2024
> Ou, Jingyang, et al. "Your absorbing discrete diffusion secretly models the conditional distributions of clean data." arXiv preprint arXiv:2406.03736 (2024).
- In the itemized list at the bottom of page 25, I believe “choose country” is repeated.

---

> ### Author Rebuttal · Authors · 2026-03-31
>
> We thank the reviewer for their detailed, thoughtful and constructive feedback. We appreciate the recognition of our contribution and the comprehensiveness of our experimental design, and are grateful for the reviewer’s suggestions which helped us identify areas for further improvement. Below we address your most important points.
>
> ### Weaknesses
>
> **Illustrating inputs in listings.** Thank you for this feedback. In the camera-ready revision, we will make sure to add a conceptual block diagram representing how the tasks are formatted, to further improve readability.
>
> **Attention Analysis.** We did indeed initially consider using attention analysis as our main tool, however, we did not observe any consistent patterns and results that would allow to explain the observed behaviours, neither in the MDLMs nor in the ARLMs. This largely agrees with previous works, which validate that in transformers gradient attribution seems to carry more signal for explaining the model behaviour (Lopardo et al. 2024, https://arxiv.org/abs/2402.03485).
>
> **Locality bias in Figure 1.** By “strong locality bias” in Figure 1, we mean that relocating relevant information within the context can change LLaDA’s accuracy by up to 20 percentage points. The big performance difference between the worst-case and the best-case scenario is what we consider a concerning threat to the robustness of the models. We will make this narrative more clear in the revised version of section 4.1.
>
> **Effect of** $1/p$ **scaling.** We find the analysis of the effect of the $1/p$ scaling an important direction of future work. Indeed, one might try training MDLMs with and without the scaling factor (or try to come up with alternative scalings) and observing the effect on the training dynamics and overall performance. However, we believe that in order for such experiments to be meaningful, they would need to consider not-small context windows (say, above 1000 tokens) and ideally focus on text data with a large vocabulary (as changing the type of token dependencies might be a confounding factor in the results). Training such models would still require considerable time, resources and warrant more detailed analysis, which we believe are outside of the scope of this work. However, we will provide an explicit mention of the importance of this kind of analysis in our section 7, when discussing future work.
>
> ### Questions
>
> **Instruct vs base models.** Prior work has demonstrated that instruction fine-tuning can deteriorate in-context few-shot learning performance in ARLMs (Irsoy et al. 2025, https://arxiv.org/pdf/2504.11626). We believe that the same effect can be observed in MDLMs.
>
> **Artefact of the synthetic task**. We agree this is an important concern and therefore include evaluations beyond the synthetic task suite. We provide some preliminary analysis in this direction in our paper already. Firstly, in Appendix A.7 we demonstrate the results on **HotPotQA**: a multi-hop reasoning dataset consisting of Wikipedia-based question-answer pairs. Results in Figure 27 showcase that also in this setting we can observe severe sensitivity to the number of masks. The observed performance trends are not as monotone as in the case of our synthetic tasks, but the results clearly show that the models are not robust to the number of masks. Also on this dataset, our mask-agnostic fine-tuning allows to alleviate this sensitivity.
>
> In Appendix A.3, we also evaluated the models on standard natural-language in-context learning tasks: AG-News, SST-2, Rotten Tomatoes, as well as MRPC, RTE and QNLI from GLUE. The results in Figure 21 demonstrate that for LLaDA models significant performance degradation occurs for a significant proportion of the datasets. Interestingly, the severity of the performance degradation is negatively correlated with how much additional in-context learning shots improve performance: such that we observe stronger performance degradation for tasks which benefit more from added context. This suggests that the observed degradation due to the extra masks occurs more generally on tasks which require long context comprehension, rather than being an artefact of the synthetic dataset used for our analysis.
>
> To further strengthen the analysis, **we have now run additional analysis on GSM8k**, as per your suggestions. We have described the details of these experiments in the response to reviewer ptCg. In short, the results demonstrate that both LLaDA and LLaDA-MoE are sensitive to the mask configurations also in this setting, with the MA loss allowing to rectify this sensitivity when trained on the GSM8k dataset (training on OpenOrca actually has a negative effect on performance).

---

> > ### Author Rebuttal · Reviewer_LHPR · 2026-04-01
> >
> > Thank you for the comprehensive response. It definitely helped clarify my questions. I especially appreciated the added result on mask degradation for a "real-world" dataset, i.e., GSM8K. As a follow-up, why do the authors think the instruct model is more sensitive to added masks and also why is there such a big dip in the 20-30 region and then a plateau beyond that?

---

> > > ### Author Response · Authors · 2026-04-08
> > >
> > > Thank you for this question — we also find this behaviour intriguing. Unfortunately, the LLaDA and LLaDA-MoE papers and associated codebases provide only limited details about the pre-training and fine-tuning procedures, which makes it difficult to precisely attribute the observed artefacts. In particular, dataset-level information (e.g., distribution of the length of question-answer pairs) that could help explain this behaviour is not available. Despite this limitation, we summarise below several observations and speculative intuitions regarding the possible origins of the observed behaviour on GSM8K.
> > >
> > > - For both LLaDA and LLaDA-MoE, the SFT stage differs from pre-training in how sequence length is handled. While pre-training operates on fixed-length sequences, SFT uses batch-dependent lengths, where shorter question–answer pairs are padded with |EOS| tokens to match the longest sequence in the batch.
> > > - Due to this preprocessing, the model never observes *fewer* masks than required to generate the true answer (it sees either the exact number or more, due to |EOS| padding). This may induce **spurious correlations** between the number of available mask positions and properties of the expected output, without masks carrying meaningful semantic information. For instance, the model may implicitly learn that the correct answer should not exceed the number of available mask positions. Importantly, this would not make masks informative in a useful sense, but could amplify the model’s sensitivity to their presence. This effect may be more pronounced in instruct‑tuned models, which are known to rely more on surface‑level heuristics and formatting cues than base models [Irsoy et al, 2025].
> > > - In the context of GSM8K, this bias may manifest as follows: when only a small number of masks is appended to the question and reasoning trace, this aligns with the model’s expectation of producing a short answer. Conversely, a large number of masks may signal the need for a longer generation (e.g., an extended reasoning trace). The intermediate regime (≈25–30 masks) may therefore be ambiguous: too many masks to safely ignore, yet insufficient to clearly trigger longer-form generation. This confusion could explain the observed dip in performance. We emphasise, however, that this remains a speculative hypothesis; access to statistics such as the distribution of sequence lengths in the SFT data would be required to validate it.
> > > - Finally, we note that across all masking regimes considered, we were consistently able to extract a valid integer prediction from the decoded outputs. In particular, the models did not degenerate into continuing the chain-of-thought instead of producing a final answer, suggesting that the degradation is due to reasoning quality rather than output formatting.
> > >
> > > While this analysis is necessarily speculative, we hope it provides useful intuition about how aspects of the SFT procedure in LLaDA and LLaDA-MoE could give rise to the observed behaviour.

---

### Official Review · Reviewer_ptCg · 2026-02-21

**Soundness:** 3
**Presentation:** 2
**Significance:** 2
**Originality:** 3
**Overall Recommendation:** 4
**Confidence:** 3

**Summary:**

This paper explores contextual understanding in Masked Diffusion Language Models (MDLMs) and identifies two key limitations. First, despite the global denoising objective of MDLMs, they exhibit strong locality bias: performance is highly dependent on the distance between relevant information and mask tokens (prioritizing nearby context over distant context), while ARLMs typically exhibit a U-shaped "lost in the middle" pattern. Second, adding additional mask tokens (necessary during generation) is distracting and degrades contextual understanding—an effect that intensifies with longer context times and is limited to mask tokens. To address these issues, the authors propose a mask-independent loss function, incorporating cross-entropy for accurate predictions and total variational distance to enforce invariance to mask counts.

**Compliance With Llm Reviewing Policy:**

Affirmed.

**Final Justification:**

accept

**Key Questions For Authors:**

(1) Will mask-independent fine-tuning affect the model's generative diversity or perplexity on standard language modeling tasks? Or is this improvement limited to contextual understanding tasks? Can some datasets be tested to support this?

(2) What is the computational cost of MA loss during training compared to standard mask diffusion training?

**Strengths And Weaknesses:**

**Strength**:

This paper not only reveals the key shortcomings (locality bias and masking interference) of the current training paradigm for MDLMs, but also provides a simple and effective fine-tuning scheme to alleviate these problems. For researchers and engineers hoping to deploy diffusion language models in practical applications, this work provides important insights and guidance.

**Weaknesses**:

(1) While the empirical evidence is strong, a more in-depth theoretical explanation is needed for why masks act as distractors (beyond the gradient attribution suggesting they attract attention). More formally linking this to the scaling factor (1/p) of the mask diffusion loss would contribute to a deeper understanding of the mechanism.

(2) The paper does not test on general datasets such as GSM8K and MMRU, making it impossible to observe the impact of fine-tuning methods on inference capabilities.

---

> ### Author Rebuttal · Authors · 2026-03-31
>
> We thank the reviewer for their thoughtful and constructive feedback. We appreciate the recognition of the practical relevance of our work, and are grateful for the reviewer’s suggestions which helped us identify areas for further improvement. Below we address your comments point by point.
>
> **W1 (analysis of** $1/p$ **scaling).** We find the analysis of the effect of the $1/p$ scaling an important direction of future work, requiring both theoretical analysis as well as its empirical validation, which taken together are outside the scope of this work. Our goal in this paper is to identify the limitations of the existing training paradigm of the MDLMs, which were not reported in previous works, and provide comprehensive empirical analysis of this phenomenon together with a potential solution. We believe this to be a necessary prerequisite both for further theoretical analysis, and for building more robust models. To encourage further research efforts, we will provide an explicit mention of the importance of further theoretical analysis of the $1/p$ scaling in our section 7, when discussing future work.
>
> **W2 (datasets).** We note that the Appendix of our paper already contains some analysis on non-synthetic datasets. In Appendix A.7 we demonstrate the results on **HotPotQA**: a multi-hop reasoning dataset consisting of Wikipedia-based question-answer pairs. Also on this dataset, our mask-agnostic fine-tuning allows to alleviate the sensitivity to mask configurations.
>
> **Additionally, we have now run the analysis on the GSM8k dataset.** To avoid introducing confounding factors, we restrict attention to a 1-step decoding setup. Following the suggestion of Reviewer LHPR, rather than requiring MDLMs to generate full reasoning chains, we prepend partial solutions (from the original dataset) to each question and task the model with generating only the final answer.
>
> We evaluate the LLaDA and LLaDA-MoE on this task.As shown in [Figure 27](https://imgbox.com/ATuT8pJy), the performance of both models—particularly the instruction-tuned variants—is sensitive to the number of masks appended to the input. Further, we evaluate the performance of LLaDA trained with the MA loss on the OpenOrca dataset (the checkpoints analysed in the main paper). We observe that while the performance flattens, it also decreases significantly overall. We hypothesise that this is because fine-tuning on an instruction-following dataset might deteriorate mathematical abilities. To further validate this claim, we then also train the LLaDA models on the GSM8k train dataset directly. As demonstrated in [Figure 28](https://imgbox.com/zR4TWf0T), this training indeed effectively makes the models invariant to the mask configurations while increasing overall performance. We further compare to a baseline training run on the GSM8k using the CE loss only and not varying the numbers of masks appended to the input ($l_1=l_2$). We observe that for LLaDA-Instruct in particular, such training does not allow to reduce sensitivity to masks.
>
> **Q1 (language modelling task).** Thank you for this suggestion: we believe such analysis will further broaden the scope of our analysis. We are currently in the process of running such analysis, and will share the results as soon as they become available (to provide us with an opportunity to share them during the rebuttal, the reviewer will have to leave a comment, such that we can then provide a final response).
>
> **Q2 (computational cost).** In order to calculate the MA loss, each training input is passed through the model twice (to account for two possible masking configurations) rather than just once.
>
> ---
>
> We hope that the introduced changes and the provided experimental results have addressed your concerns and if so, we would be grateful if you would consider reflecting this in your score. We are happy to discuss any further questions.

---

> > ### Author Rebuttal · Reviewer_ptCg · 2026-04-02
> >
> > Thank you for the response.

---

> > > ### Author Response · Authors · 2026-04-03
> > >
> > > Thank you for giving us the opportunity to run additional experiments into the effect of MA loss on language modelling capabilities of MDLMs. As promised, we present the results below.
> > >
> > > **Experimental Setup.** We evaluate the language modelling performance on the [Wikitext](https://huggingface.co/datasets/Salesforce/wikitext) and [LM1B](https://huggingface.co/datasets/billion-word-benchmark/lm1b) datasets (standard language modelling datasets, that were not used for training using the MA loss). To ensure a fair comparison across all models, we consider a setting aligned with the training objective of MDLMs, namely *token infilling under heavy masking*. For each text sample, we first sample the masking probability uniformly from the interval [0.6, 0.9], and then independently mask input tokens according to this probability. This results in inputs with 60–90% masked tokens (on average) at random positions, requiring substantial reconstruction of global context. We then perform iterative unmasking using the *high-confidence decoding* scheme, unmasking one token at a time. We evaluate on 250 samples per dataset, restricting sequence length to 100–400 tokens to control computational cost. We report two complementary metrics: ROUGE-L (measuring the longest common subsequence between the generated text and a reference) and Generative PPL (calculated as the perplexity of a reference model, in our case Qwen2.5-Base-8B, on the text generated by the model being evaluated). This combination allows us to jointly assess *faithfulness* (ROUGE-L) and *linguistic plausibility* (Gen-PPL).
> > >
> > > **Results.** The results, presented in [this table](https://imgbox.com/ZsR3tTB4), demonstrate that the MA loss has little to no significant effect on the language modelling capabilities of the models. These findings suggest that incorporating the MA loss does not adversely affect the language modelling behaviour of MDLMs in high-mask infilling regimes. This is consistent with the formulation of the objective, which retains a cross-entropy component that anchors the model to the underlying token distribution.
> > >
> > > ---
> > > We hope that these results resolve your outstanding concerns and as a result, you will consider increasing your score.

---

### Official Review · Reviewer_kBcX · 2026-02-25

**Soundness:** 4
**Presentation:** 3
**Significance:** 4
**Originality:** 4
**Overall Recommendation:** 6
**Confidence:** 3

**Summary:**

This paper investigates how masked diffusion language models (MDLMs) use context, focusing on positional sensitivity and the effect of appending mask tokens that are commonly used for generation. Through controlled few-shot multiple-choice style tasks, the authors find that MDLMs exhibit a locality bias similar in spirit to ARLM recency effects, and that adding many mask tokens can sharply degrade accuracy for models trained from scratch (notably LLaDA). To mitigate this, the paper proposes a mask-agnostic fine-tuning objective combining cross-entropy with a distribution-matching term (TV distance) to encourage invariance to the number of appended masks, improving robustness and partially reducing locality effects.

**Compliance With Llm Reviewing Policy:**

Affirmed.

**Final Justification:**

I believe this paper meets the bar for acceptance at ICML, and therefore I will maintain my positive assessment.

**Key Questions For Authors:**

If you can reasonably address my concerns and provide a discussion on the stability of the results during the rebuttal phase, I would be more than happy to increase my score.

**Limitations:**

yes

**Strengths And Weaknesses:**

Strengths.
1. Identification of Novel Phenomena: The paper systematically identifies two counter-intuitive phenomena in MDLMs: first, despite a global denoising objective, these models exhibit a strong "locality bias," relying heavily on context adjacent to the masks. Second, in MDLMs trained from scratch (e.g., LLaDA), increasing the number of mask tokens leads to a significant and monotonic performance degradation (an "inverse scaling law").
2. Rigorous Ablation Analysis: The authors designed controlled experiments to validate their hypotheses. For instance, by replacing extra masks with standard dots ("."), they successfully ruled out the possibility that the performance drop was merely due to "repetitive tokens," proving that the mask tokens themselves act as strong distractors. Furthermore, gradient attribution analysis confirms at a mechanistic level that the models allocate disproportionate attention to these masks.
3. Actionable Solution: The paper goes beyond identifying the problem by proposing a "Mask-Agnostic (MA)" loss function. This approach combines Cross-Entropy (CE) and Total Variation (TV) loss to align probability distributions across different mask configurations. Experiments demonstrate that this fine-tuning method effectively improves model robustness to mask counts and reduces locality bias while maintaining low decoding latency.

Weaknesses. I am not a domain expert in this sub-area, so I do not attempt to judge the paper's novelty; instead, I focus on the empirical evaluation and reproducibility.

1. Ambiguity/possible error in MA-loss normalization and notation. The MA loss in Section 6.1 is described with sufficient mathematical detail to reimplement in principle, but the presentation has a few confusing points: (n_m) is defined as (n_m = \sum_{j \in \mathcal{A}} \mathbb{1}{x_i^j = m}) while (i\in{1,2}). As written, this depends on (i), yet you use the same (n_m) for both (i=1,2) in (\mathcal{L}{CE}), and also in (\mathcal{L}{TV}) which only gates on (x_1). This ambiguity matters because it changes the effective weighting when (l_1 \neq l_2) and/or if padding differs. You likely intend (n_m) to mean the number of masked answer tokens in (\tilde a) (which is shared across both inputs), not including appended masks. If so, please rewrite it explicitly.
2. Generalization beyond LLaDA is not convincingly established in the main body. The strongest “mask distraction” effects are for LLaDA (Figure 3), while Dream is much more robust and even shows similar effects for dots vs masks (Figure 5). This undermines the paper’s broad title/abstract framing unless the claims are narrowed or further supported with additional MDLM families in the main text.
3. MA fine-tuning controls are not tight enough to isolate causal factors. Different learning rates/(\beta) settings (Appendix B) and the lack of a “format-matched but invariance-free” control mean it is still unclear how much of Figure 8’s gain is due specifically to mask-count invariance rather than generic adaptation.
4. Decoding-vs-masks confounding in the “mask tax” discussion. Figure 6 shows multi-step unmasking largely recovers performance, which suggests the severity of the problem depends strongly on the decoding regime. Section 7 frames a broad deployment friction point, but the paper does not clearly delineate which real-world decoding setups are most impacted.

Suggestions.
1. Key mechanistic evidence is relegated to appendix. Table 1 is directly relevant to the “distractor” mechanism but is not integrated into the main narrative.

---

> ### Author Rebuttal · Authors · 2026-03-31
>
> We thank the reviewer for their thoughtful and constructive feedback. We appreciate the recognition of our contribution and the rigour of our experimental design, and are grateful for the reviewer’s suggestions which helped us identify areas for further improvement. Below we address all your comments point by point.
>
> ### Weaknesses
>
> **W1.** Thank you for this detailed question. We note that in our notation $n_m$ is defined as a sum over $j \in \mathcal{A}$, where in line 370-371 we define $\mathcal{A}$ as the set of indices of the elements of $\mathbf{x}_1$ and $\mathbf{x}_2$ which correspond to the answer-part of the input. Therefore, we count only the masks in $\tilde{\mathbf{a}}$, which is shared between $\mathbf{x}_1$ and $\mathbf{x}_2$ (similarly, the loss is computed only over the mask tokens in $\tilde{\mathbf{a}}$). We recognise, however, that this might have been confusing, and in the camera-ready version we will make sure to clarify that we are counting the masks only present in $\tilde{\mathbf{a}}$.
>
> **W2.** Thank you for this suggestion. In the camera-ready version of the paper, we will use the extra page available to move at least some of the results on LLaDA-MoE (currently in Appendix A.1) to the main paper, this way showcasing the generality of our findings.
>
> **W3.** Our choice of the hyperparameters for the controls was guided by trying to optimise the performance of each individual fine-tuning scheme, thus allowing for a more favourable performance of the baseline loss functions. This experimental design choice was guided by the fact that the TV loss and the CE loss tend to operate at different scales (orders of magnitude), and thus might require different scaling factors $\alpha, \beta$ and learning rates. However, to further improve the robustness of our comparison we have now run two additional controls using the same hyperparameters as the MA loss (lr=1e-5 for the Base model).
>
> 1. **CE only:** Fine-tuning with CE loss only, at the same learning rate as CE+TV — this directly rules out the learning rate as a confounding factor.
> 2. **CE with equal lengths:** Fine-tuning with CE loss at the same learning rate, but using a single fixed mask length rather than sampling two different lengths.
>
> The results, shown in [Figure 32](https://imgbox.com/Vkh7V8n8), demonstrate that both controls (when trained with the same learning rate) fail to maintain accuracy as the number of mask tokens increases. By contrast, CE+TV (MA loss) maintains high accuracy while achieving robustness to mask count. This demonstrates that the TV invariance term is a critical ingredient: neither a higher learning rate nor exposure to varied mask lengths during training is sufficient to achieve the gains observed with the MA loss.
>
> **W4.** Thank you for bringing up this point. In the revised version of the paper, we will explain that the “mask tax” applies primarily when decoding in few steps (see Appendix A.4 for results showing that mask-agnostic training is also beneficial when using 2, 4, 6 decoding steps) and can be largely avoided by decoding one token at a time (as we demonstrate in Figure 6). Few-step decoding is a setting particularly relevant in low-latency applications, when trying to minimise the computational cost and sampling time; it is also relevant when attempting to perform adaptive parallel decoding (Israel et al., 2025, https://arxiv.org/pdf/2506.00413), where the goal is to sample as many tokens in parallel as possible.
>
> **Suggestions.** Thank you for your suggestion, we will aim to move Table 1 to the main paper in the camera-ready version.
>
> ---
>
> We hope the proposed changes, additional results and provided clarifications addressed your concerns and if so, we would be grateful if you would consider reflecting this in your score. We are happy to discuss any further questions.

---

> > ### Author Rebuttal · Reviewer_kBcX · 2026-04-01
> >
> > Thank you for the response. Although I am not an expert in this field, I sincerely appreciate your genuine responses to my questions. I would be very grateful if you could incorporate our suggestions in the revised version. In line with my commitment during the rebuttal process, I have decided to raise my score. Congratulations, and I hope your paper will be accepted successfully.

---

### Official Review · Reviewer_14Ne · 2026-02-25

**Soundness:** 3
**Presentation:** 2
**Significance:** 3
**Originality:** 3
**Overall Recommendation:** 4
**Confidence:** 3

**Summary:**

The paper investigates the context comprehension abilities of Masked Diffusion Language Models (MDLMs) and presents two central findings: (1) despite their global denoising objective, MDLMs still exhibit a pronounced locality bias; and (2) appending additional mask tokens at inference time can significantly degrade performance.

**Compliance With Llm Reviewing Policy:**

Affirmed.

**Key Questions For Authors:**

1. The paper lacks interpretability-oriented analyses, such as attention analysis, entropy analysis, and gradient-based investigations, which makes the current study feel incomplete.

2. The experimental evaluation is not sufficiently comprehensive; more diverse and rigorous benchmarks are needed to better assess the effectiveness of the approach.

3. I encourage the authors to expand the discussion on diffusion-based MLLMs, for example models such as LLaDA-V.

4. The experimental analysis regarding mask tokens should be more thorough and systematically examined.

5. I understand that the rebuttal timeline is limited, but I would appreciate more insightful discussion from the authors, particularly on broader implications and future research directions.

**Limitations:**

yes

**Strengths And Weaknesses:**

This work investigates a practically relevant and well-motivated problem. The overall experimental effort is substantial; however, several important experiments are still missing. If the authors can address these issues during the rebuttal stage, I would consider revising my score upward.

---

> ### Author Rebuttal · Authors · 2026-03-31
>
> We thank the reviewer for their feedback. We appreciate the recognition of our contribution, the practical relevance of the problems we study, and the extensiveness of our experimental results. Below we answer all of your questions.
>
> **Question 1:** We agree with the reviewer that interpretability-oriented analyses are important towards improving the robustness and completeness of our results. For this reason, **in the Appendix of the submitted paper, we already present the results of such an analysis**. Specifically:
>
> 1. Appendix A.2 (mentioned in lines 158-160) demonstrates the results of a gradient-attribution analysis on the MDLMs and ARLMs, which provides further mechanistic evidence for the observed locality biases. It further shows that indeed, the MDLMs are particularly sensitive to the mask tokens, as opposed to other tokens in the input. We decided to use gradient attribution analysis (rather than attention analysis) as it has been shown by prior work to carry more signal for explaining the model behaviour (Lopardo et al. 2024, https://arxiv.org/abs/2402.03485).
> 2. Appendix A.6 showcases how the confidence and entropy of the model in the generated predictions both change as a function of masks. The results demonstrate that both the confidence and the entropy of the model are sensitive to the number of masks generated. This effect can be alleviated by fine-tuning the model with the MA-loss.
>
> In the camera-ready version of this paper, we will make sure to include at least some of the gradient-attribution analysis results in the main paper, using the extra page allowed.
>
> **Question 3:** We note that the current paper focuses on purely text-based models, rather than on multimodal models such as LLaDA-V.
>
> **Question 2&4:** We encourage the reviewer to further review the contents of **our Appendix, and the experiments presented within**, which further explore the role and effect of mask tokens, across different benchmarks:
>
> 1. Results on additional datasets, including HotPotQA (a multi-hop reasoning dataset) in Appendix A.7 and a synthetic multidimensional-classification dataset in Appendix A.8.
> 2. The study of the correlation between mask degradation and context significance in Appendix A.3, which includes evaluation on many different in-context learning datasets including AG News, SST-2, Rotten Tomatoes, as well as MRPC, RTE and QNLI from GLUE. The results show that for the LLaDA models, the performance decrease caused by extra masks is stronger on tasks which require more context processing.
> 3. The study of the confidence and entropy as a function of the extra masks in Appendix A.6.
> 4. Results showing that our MA-loss improves performance also when decoding in few decoding steps (2, 4, 6) in Appendix A.4.
>
> Additionally, we have now conducted an analysis on the GSM8k dataset (see response to reviewer ptCg). *Should the reviewer have any ideas for outstanding experiments which are crucial and would substantially improve the robustness of the presented results, we will gladly take such concrete suggestions into consideration.*
>
> **Question 5:** Thank you for this suggestions. Section 7 of our paper already presents some implications of our work which we believe warrant a discussion: the general problem of “mask tax” on parallel decoding, and the resulting MDLM evaluation guidelines. In the same section we also identify the examination of uniform (rather than masked) discrete diffusion models as an area for future work. If we were to propose additional directions for future work, we would consider the following:
>
> 1. Exploring empirically whether the identified locality bias of MDLMs can be alleviated by carefully tuning the $1/p$ scaling in the MDLM training objective during training (cf. lines 201-210).
> 2. Evaluating the effect of training models from scratch using the MA-loss (rather than only fine-tuning), which, however, requires significant computational resources.
>
> ---
>
> We hope these clarifications addressed your concerns and if so, we would be grateful if you would consider reflecting this in your score. We are happy to discuss any further questions.

---

> > ### Author Rebuttal · Reviewer_14Ne · 2026-04-01
> >
> > Thank you for the response.

---

> > > ### Author Response · Authors · 2026-04-08
> > >
> > > Thank you again for the thoughtful review; we're glad that our rebuttal fully resolved your concerns.
> > >
> > > We noticed, however, that your overall score has not yet been updated from your original score in your review. We wanted to send a quick follow-up to make sure that this wasn't a system bug or a simple oversight.
> > >
> > > We further note that in the meantime we ran some additional analysis about the effect of our mask-agnostic loss on the language modelling performance of LLaDA and LLaDA-MoE: you can see the results and discussion in a comment to reviewer ptCg. We hope this helps to further strengthen the experimental evaluation of the paper.

---

### Decision · Program_Chairs · 2026-04-30

**Decision:**

Accept (regular)

**Comment:**

This paper investigates the context comprehension capabilities of Masked Diffusion Language Models (MDLMs), identifying two critical limitations: (i) a locality bias and (ii) a performance degradation effect caused by appended mask tokens, termed "mask distraction." Reviewers appreciated the identification of these phenomena and the ablation studies used to rule out confounding factors like token repetition. While initial concerns were raised regarding the synthetic nature of the task and interpretability, the authors provided a detailed rebuttal that included new experiments on GSM8K, HotPotQA, and standard natural-language benchmarks. Furthermore, the authors clarified the formulation of their proposed mask-agnostic (MA) loss and utilized gradient-attribution analysis to provide deeper mechanistic evidence for the model's behavior. These additions resolved concerns regarding generalizability and technical clarity, resulting in a unanimous positive consensus among the reviewers. Given the importance of these findings for the future design and evaluation of diffusion-based language models, the paper is recommended for acceptance.